

# Precipitation δ¹⁸O on the Himalaya-Tibet orogeny and its relationship to surface elevation

Hong Shen[1], Christopher J. Poulsen[1]

[1]Department of Earth and Environmental Sciences, University of Michigan, Ann Arbor, 48109, USA

5   *Correspondence to*: Hong Shen (hdshen@umich.edu)

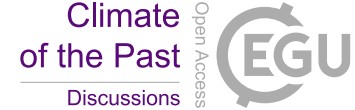

**Abstract.** The elevation history of the Himalaya-Tibet orogen is central to understanding the evolution and dynamics of both the Indian-Asia collision and the Asian monsoons. The surface elevation history of the region is largely deduced from stable isotope ($\delta^{18}O$, $\delta D$) paleoaltimetry. This method is based on the observed relationship between the isotopic composition of meteoric waters ($\delta^{18}O_p$, $\delta D_p$) and surface elevation, and the assumption that precipitation undergoes Rayleigh distillation under forced ascent. Here we evaluate how elevation-induced climate change influences the $\delta^{18}O_p$-elevation relationship and whether Rayleigh distillation is the dominant process affecting $\delta^{18}O_p$. We use an isotope-enabled climate model, ECHAM-wiso, to show that the Rayleigh distillation process is only dominant in the monsoonal regions of the Himalayas when the mountains are high. When the orogen is lowered, local surface recycling and convective processes become important as forced ascent is weakened due to weaker Asian monsoons. As a result, the $\delta^{18}O_p$ lapse rate in the Himalayas increases from around −3 ‰/km to above −0.1 ‰/km, having little relationship with elevation. On the Tibetan Plateau, the meridional gradient of $\delta^{18}O$ decreases from ~1 ‰/° to ~0.3 ‰/° with reduced elevation, primarily due to enhanced sub-cloud re-evaporation under lower relative humidity. Overall, we report that using $\delta^{18}O_p$ or $\delta D_p$ to deduce surface elevation change in the Himalaya-Tibet region has severe limitations and demonstrate that the processes that control $\delta^{18}O_p$ vary by region and with surface elevation. In sum, we determine that the application of $\delta^{18}O$-paleoaltimetry is only appropriate for 7 of the 50 sites from which $\delta^{18}O$ records have been used to infer past elevations.

## 1 Introduction

Surface elevation is a fundamental characteristic of the Earth surface, directly affecting atmospheric circulation patterns and surface temperatures, precipitation and surface hydrology, erosion and sediment transport, and the distribution and diversity of life (Aron and Poulsen, 2018). The evolution of surface elevation resulting from India-Asia convergence and creation of the Himalaya-Tibet Orogen was the defining event of the Asian continent in the Cenozoic and had global environmental implications. Surface uplift of the orogen has been implicated in the onset and strengthening of the southeast Asian monsoon system (Boos and Kuang, 2010; Zhang et al., 2015) and the position of atmospheric stationary waves (Kutzbach et al., 1989); the evolutionary diversification and biogeographic distribution of fauna and flora in central Asia (Zhao et al., 2016; Yang et al., 2009); and, the intensification of chemical weathering of exposed rocks and transport of nutrients to the ocean that contributed to global atmospheric $CO_2$ drawdown (Raymo et al., 1988). Surface elevation also lends a first-order constraint on the crustal and upper mantle dynamics that create topography (Ehlers and Poulsen, 2009). Surface elevation estimates for the Himalaya-Tibet orogen, and specifically evidence for high elevations of the orogen since the late Eocene, have been instrumental in supporting geodynamical models of Tibetan Plateau growth through early deformation and crustal thickening (Rowley and Currie, 2006; Rohrmann et al., 2012; Hoke et al., 2014).

The paramount importance of surface elevation to our understanding of Cenozoic environmental and tectonic evolution has led to a proliferation of studies that infer past Himalaya-Tibet surface elevations from ancient proxy materials (Cyr et al.,





2005; Rowley and Currie, 2006; Li et al., 2015). Stable isotope paleoaltimetry, one of the few quantitative methods to reconstruct past surface elevations, relies on the water isotopic composition of ancient materials, including pedogenic carbonate, authigenic clay and hydrated volcanic glass, that were formed in contact with ancient surface waters. The method is predicated on the observed, modern decrease in water stable isotopic compositions ($\delta^{18}O$, $\delta D$) with elevation gain (the

isotopic lapse rate) (Chamberlain and Poage, 2000) a relationship that is commonly attributed to rainout of heavy isotopologues during stably forced ascent of a saturated air over high elevation and is modeled as a Rayleigh distillation process (e.g. Rowley and Garzione, 2007). Stable isotope paleoaltimetry has been used to reconstruct past surface elevation of many of the world's major mountain belts, including the North America Cordillera (e.g. Poage and Chamberlain, 2002; Fan et al., 2014), the Andes (e.g. Garzione et al., 2008), and the Himalayan-Tibetan orogen (e.g. Rowley and Currie, 2006),

due to the robustness of the isotopic lapse rate in modern orogenic regions and the ubiquity of proxy materials.

The interpretation of stable isotopes in ancient materials to infer past surface elevation is complicated by factors related both to the mineralization of proxy materials and to the isotope-elevation relationship of meteoric waters from which the proxies form (Poage and Chamberlain, 2001). With regard to the latter factor, studies using global climate models have demonstrated that the isotopic lapse rate can be dependent on a mountain range's elevation due to processes that are not described by

Rayleigh distillation (Ehlers and Poulsen, 2009; Poulsen et al., 2010; Feng et al., 2013; Botsyun et al., 2016). Indeed, Feng et al. (2013) showed that the simulated $\delta^{18}O$ of precipitation ($\delta^{18}O_p$) during uplift of the Eocene North America Cordillera was substantially influenced by changes in vapor mixing, surface recycling, moisture source change, and precipitation type. Similarly, Botsyun et al. (2016) investigated $\delta^{18}O_p$ across the Himalaya-Tibet region in response to surface uplift and showed that direct topographic effects only partially accounted for total $\delta^{18}O_p$ changes.

On the Himalayan slope, $\delta^{18}O$ in surface waters and precipitation has been widely observed to decrease with elevation at a rate of ~3 ‰/km (e.g. Rowley et al., 2001). The $\delta^{18}O$-elevation relationship has been attributed to orographic rainout and modeled as a Rayleigh distillation process (e.g. Rowley and Currie, 2006). On the Tibetan plateau, $\delta^{18}O$ in surface water and precipitation increases linearly with latitude by ~1 ‰/° over nearly uniform elevation (e.g. Bershaw et al., 2012). The source of $\delta^{18}O$ variations on the plateau and whether $\delta^{18}O$ can be used for paleoaltimetry on the high Tibetan Plateau has received

little attention. Bershaw et al. (2012) proposed that paleoelevations could be inferred from proxy $\delta^{18}O$ after removing the meridional $\delta^{18}O$ gradient on the Tibetan Plateau. However, little is known about either the processes that contribute to the meridional $\delta^{18}O$ gradient or how the meridional $\delta^{18}O$ gradient varied when the plateau was lower.

The goal of this study is to identify and quantify the processes that control $\delta^{18}O_p$ variations across the Himalayas and the Tibetan Plateau and to evaluate the utility of $\delta^{18}O_p$ as a paleoaltimeter in these regions. To do this, we use an isotope-enabled

global climate model, ECHAM5-wiso, with prescribed elevation scenarios and compare the $\delta^{18}O_p$ calculated by the climate model with that expected due to Rayleigh distillation alone. Botsyun et al. (2016) used the LMDZ-iso model to decompose the influence of adiabatic elevation changes on $\delta^{18}O_p$ from other influences due to non-adiabatic temperature changes, local changes in relative humidity and post-condensational processes. Building on Botsyun et al. (2016), we take a more process-





oriented approach to quantify the isotopic fluxes attributed to specific mechanisms and demonstrate that the contributions from these processes vary spatially and in response to elevation change. Finally, we discuss the implications of our results for reconstructing paleoaltimetry of the Himalayas and Tibetan Plateau.

## 2 Methods

### 2.1 Model and experimental design

In this study, we employ ECHAM5-wiso, a water isotope-enabled atmospheric global climate model (AGCM). The model has been widely used both for modern and past climate and isotope simulations. For instance, Feng et al. (2013) and Feng and Poulsen (2016) employed ECHAM5-wiso to explore climate and isotopic responses to Cenozoic surface uplift and climate change in western North America. ECHAM has also been shown to simulate many aspects of Asian climate (e.g. Battisti et al., 2014) and isotopic compositions ($\delta^{18}O_p$) simulated by ECHAM5-wiso generally agree well with observed modern stream and precipitation $\delta^{18}O$ across the Tibetan Plateau as shown by Li et al. (2016) (see their Fig. 11).

We use a model configuration with 19 vertical levels, and a spectral triangular truncation of 106 horizontal waves, approximately equivalent to a 100-km grid spacing. This horizontal resolution, though still relatively coarse, is about twice that of recent simulations used to validate the simulation of water isotopes over the Tibetan plateau (Li et al., 2016) and recent paleoclimate simulations of the Tibetan Plateau (e.g. Roe et al., 2016). The AGCM is coupled to a slab-ocean model, the MPI-OM, with prescribed monthly sea surface temperature and ocean heat flux from the Atmospheric Modeling Project Intercomparison 2 (AMIP2; Gleckler, 2005) averaged over years 1956-2000. A modern seawater $\delta^{18}O$ dataset spanning from 1956 to 2006 (LeGrande and Schmidt, 2006) is provided as a lower boundary condition for the model. In ECHAM5-wiso, water isotopologues are included as independent tracers in the atmosphere. When water evaporates from the sea, both equilibrium and non-equilibrium distillation processes occur as a function of sea surface temperature, wind speed, relative humidity, isotope composition in seawater and vapor above the ocean surface (Hoffmann et al., 1998). Convective rains are assumed to have large rain drops that reach only partial (50%) isotopic equilibration with surrounding vapor, while large-scale precipitation with smaller rain drops attains almost complete (90%) equilibration with the environment (Hoffmann et al., 1998). Only large lakes, the size of at least one-half grid cell, are resolved in the model, and fractionation from the land surface is not included since its impact on precipitation $\delta^{18}O_p$ is negligible (Haese et al., 2013).

We conducted two sets of sensitivity experiments (Fig. 1) in addition to a control simulation with modern conditions (CNTL). In the first set of sensitivity experiments, topography is uniformly lowered to 80% (TOPO80), 60% (TOPO60), 40% (TOPO40) and 20% (TOPO20) of its modern elevation over a domain that includes the Himalaya and Tibetan plateau. In the second set of sensitivity experiments, we conducted two experiments with non-uniform elevation modifications over this domain. The first experiment includes a high Himalayan front with a Tibetan Plateau reduced to 20% of its modern elevation (TOPO20a). This experiment is inspired by the widely accepted notion that southeast Asia had an Andean type



mountain belt before the collision of the Eurasia and the Indian plate (Royden et al., 2008). TOPO20a also serves as a test of the Himalaya on the regional climate. In the second experiment, the outer edge of the Tibetan Plateau remains, but the inside is lowered to 20% of its modern height (TOPO20b). The second experiment is a sensitivity test to investigate the role of plateau heating on regional climate and isotopic compositions. In both sets of experiments, we tapered the topography along

the borders of the domains to avoid any abrupt topography boundaries. Except for topography, all other boundary conditions are kept the same among all experiments. Each experiment was run for 20 years, with the last 15 years used for analysis. Only summertime (June-July-August) climate variables and precipitation-weighted $\delta^{18}O$ are analyzed, since carbonates form primarily under summer temperature when precipitation peaks (Peters et al., 2013), and both summer climate and precipitation-weighted $\delta^{18}O$ are most commonly reconstructed for paleoclimate studies (e.g. Quade et al., 2011; Bershaw et

al., 2012).

Mean climate conditions today vary across the Himalaya; the western Himalaya is characterized by peak precipitation in winter and early spring, while the central Himalaya is dominated by the Indian Summer Monsoon (IM) and the eastern Himalaya by the East Asia Summer Monsoon (EASM) (Yao et al., 2013). Because of this heterogeneity, we separate the Himalayas into four distinct regions for analysis purposes, including the western Himalaya; a transitional area between

western and central Himalaya; central Himalaya; and eastern Himalaya (transitional areas between central and eastern Himalaya are excluded because climate and isotopic signals are similar to IM and EASM regions). The strength and pattern of precipitation and wind, and isotopic compositions in the two eastern-most transitional areas are similar to those in the IM and EASM in most cases. Thus, in the following we present results only for the western, transitional, IM and EASM regions as shown in Fig. 2.

**2.2 Rayleigh distillation**

We developed an open-system, one-dimensional, altitude-dependent Rayleigh distillation model (RDM) in order to estimate decreases in $\delta^{18}O_p$ due to Rayleigh distillation during ascent. The RDM tracks the isotopic composition of an air parcel as it ascends adiabatically from low to high altitude, becomes saturated, and loses condensate through precipitation. In the RDM, an air parcel cools at the dry adiabatic lapse rate before condensation and at the moist adiabatic lapse rate upon saturation

(Rowley and Garzione, 2007). The RDM is run using terrain-following coordinates and is initialized with three different moisture sources: (1) fixed air temperature (T=20°C) and relative humidity (RH=80%), (2) local, low-level T and RH from ECHAM5, and (3) fixed T=20°C and RH from ECHAM5. In this way, we are able to quantify the influence due to total moisture source change ((1) minus (2)), and further decompose this influence into the changes in T ((3) minus (2)) or RH ((1) minus (3)).

In order to estimate how much of the mass flux of $^{18}O$ in total precipitation is contributed by the Rayleigh distillation process, we assumed that all large-scale precipitation, $P_l$, forms in response to stable upslope ascent and participates in Rayleigh distillation. We then estimated the isotopic flux of water undergoing Rayleigh distillation as:





$$RD = \left(\frac{\delta^{18}O_{RDM}}{1000} + 1\right) R_{vsmow} \times P_l \times \rho_{water}, \tag{1}$$

where $RD$ has units of g m$^{-2}$ h$^{-1}$; $\rho_{water}$ (in g m$^{-3}$) is the density of water; $P_l$ (in m h$^{-1}$) is the large-scale precipitation rate; $\delta^{18}O_{RDM}$ is the isotopic composition simulated by the RDM at the same elevation as the grid points in 2.3 where the mass flux in total precipitation is estimated. Note that this estimation of $RD$ stands for the upper limit of the contribution of $RD$,

since not all large-scale precipitation is triggered by the Rayleigh distillation process.

To compare the relative importance of Rayleigh distillation with other isotopic fractionation processes in ECHAM, we quantify the change in upslope $\delta^{18}O_p$ attributable to Rayleigh distillation. We do this by comparing the rate of upslope $\delta^{18}O_p$ change (hereafter referred to as the $\delta^{18}O_p$ lapse rates) in ECHAM with that estimated using the RDM. We define the $\delta^{18}O_p$ lapse rate as the slope of the linear regression equation of precipitation-weighted $\delta^{18}O_p$ regressed on elevation and use the

coefficient of determination ($R^2$) of this regression to evaluate the robustness of the $\delta^{18}O_p$-elevation relationship. We use a ratio of the $\delta^{18}O_p$ lapse rate in ECHAM to that in the RDM (p_percent) to approximate the contribution of Rayleigh distillation to the ECHAM $\delta^{18}O_p$ lapse rate. When $R^2$ and p_percent are both above 0.5 for a particular domain and elevation scenario, we consider the ECHAM $\delta^{18}O_p$ lapse rate to agree with the RDM $\delta^{18}O_p$ lapse rate.

To evaluate whether a strong $\delta^{18}O_p$-elevation relationship exists in low-elevation scenarios, we examine the ratio of the slope

of $\delta^{18}O_p$ regressed against latitude on the subcontinent to the south of the Himalayas to that on the Himalayan slope. This method accounts for the fact that $\delta^{18}O_p$ changes with both latitude and elevation. A significant $\delta^{18}O_p$-elevation relationship is signified by a large ratio, indicating that $\Delta\delta^{18}O_p$ with elevation is greater than that with latitude.

### 2.3 Quantifying effects through the mass flux of $^{18}O$

Mass fluxes of $^{18}O$ are calculated to quantify the contribution of vapor mixing, surface recycling and RDM to the $^{18}O$ in total

precipitation. Vapor mixing and surface recycling serve as the lateral and lower boundary sources of $^{18}O$ in an air column, and total precipitation as the sink. Within this air column, sinks and sources of $^{18}O$ in the air should compensate to make the total $^{18}O$ in the air stable at a constant value in the long term on climate time scales. This relationship of sources and sinks indicates that the mass flux of $^{18}O$ from vapor mixing and surface recycling should approximate that from total precipitation. With this compensation of sources and sinks in mind, we calculated the flux of $^{18}O$ due to vapor mixing, surface recycling

and total precipitation.

The vapor mixing between air masses is estimated as the advection of $^{18}O$ in a vertical air column:

$$\frac{\partial M^{18}O}{\partial t} \sim \int -\vec{V} \cdot \vec{\nabla} m^{18}O \, dz, \tag{2}$$

where $M^{18}O$ (g m$^{-2}$) is the unit column-total mass of $^{18}O$ in the air; $\vec{V}$ (m h$^{-1}$) is the wind speed vector within a layer; $m^{18}O$ (g m$^{-3}$) is the mass of $^{18}O$ per m$^3$ of air, and z (m) is elevation. Approximating total mass of water by the amount of $^{16}O$,

$m^{18}O$ is defined in terms of $\delta^{18}O_c$ from (1) as:





$$m^{18}O \approx q\rho_{air}\left(\frac{\delta^{18}O}{1000} + 1\right)R_{vsmow} \ . \tag{3}$$

By substituting $\delta^{18}O_c$ in (3) into (2), the final form of the total column-integrated vapor mixing (g m$^{-2}$ h$^{-1}$) is written as:

$$VM = \int -\vec{V} \cdot \vec{\nabla} m^{18}O \ dz = -\int \vec{V} \cdot \vec{\nabla}\left[q\rho_{air}\left(\frac{\delta^{18}O}{1000} + 1\right)R_{vsmow}\right]\rho_{air}dz \ .$$
$$\tag{4}$$

Note that the centered-finite-difference method is used in discretizing the derivatives in Eq. (4). This method could potentially bring errors in comparison to the spectral method used in the dynamical core of ECHAM5.

Recycling of surface water vapor transports $^{18}$O to the atmosphere from lower boundary. To estimate this contribution to $^{18}$O, the recycled mass flux (in g m$^{-2}$ h$^{-1}$) is calculated as:

$$\left(\frac{\delta^{18}O}{1000} + 1\right)R_{vsmow} \times E \times \rho_{water}, \tag{5}$$

where $\delta^{18}O$ is the isotopic composition of the evaporated water and $E$ (m h$^{-1}$) is the surface evaporation rate.

The mass flux of $^{18}O$ in total precipitation is estimated following (3) as:

$$\left(\frac{\delta^{18}O}{1000} + 1\right)R_{vsmow} \times P \times \rho_{water}, \tag{6}$$

where $\delta^{18}$O is the isotopic composition of precipitation and $P$ (m h$^{-1}$) is the precipitation rate.

## 3 Results

### 3.1 Climate response to Tibetan-Himalayan surface elevation

Numerous modeling studies have found that lowering the height of the Tibetan Plateau influences regional temperature, wind, precipitation, and relative humidity (e.g, Kitoh, 2004; Jiang et al., 2008). In this section, we describe regional climate changes on the western Himalayan slope, the Tibetan Plateau, the IM region and the EASM region, that occur as high elevations are lowered.

Under modern conditions, near-surface temperatures in the Tibetan-Himalayan region vary with elevation following a moist adiabatic lapse rate (~5 °C km$^{-1}$) (Fig. S1) and range from >30 °C on the Indian subcontinent at the foot of the Himalaya to <10 °C across the Tibetan Plateau (Fig. S2a). Lowering elevations in our experiments causes near-surface temperatures across the Tibet-Himalayan region to increase (Fig. S2b-g) at approximately the lapse rate for CNTL. As a result, temperature lapse rates vary little among elevation scenarios, for instance, ranging from 4.9-5.4 °C km$^{-1}$ in the IM region

(Fig. 1).

Wind patterns, precipitation and RH respond dramatically to reductions in elevation. These changes vary across regions. On the western Himalayan slope, wind directions nearly reverse with southerly winds in the high-elevation scenarios switching to northwesterly winds in low-elevation scenarios (Fig. 3). This wind reversal results in transport of arid air from the north, lowering total-column relative humidity by ~40%. With this substantial decrease in RH, summer precipitation decreases

from ~1100 mm yr$^{-1}$ in CNTL to ~30 mm yr$^{-1}$ in TOPO20. A similar decrease in RH, by ~20% from CNTL to TOPO20, is



simulated on the Tibetan Plateau. Accompanying this reduction in RH, precipitation decreases from ~1300 mm yr$^{-1}$ in CNTL to ~500 mm yr$^{-1}$ in TOPO20. This reduction in Tibetan Plateau precipitation is linked to a weakening of the Asian monsoonal systems and moisture delivery through monsoonal winds.

The responses in the monsoonal regions are somewhat different from those on the western slope and Tibetan Plateau. A reduction in surface elevation (from CNTL to TOPO20) (Fig. 1) leads to weakening of the IM, as indicated by slowing of summer southwesterly winds over the Bay of Bengal and the Arabian Sea and a decrease in summer precipitation (by more than 20 mm day$^{-1}$ in ECHAM) along the central Himalayas (Fig. 3). IM weakening is also demonstrated by the WSI1 monsoon index (Fig. 4, following Wang and Fan (1999)), which is defined by the vertical wind shear between the lower (850 hPa) and upper (200 hPa) troposphere in the region (5-20° N, 40-80° E). Monthly WSI1 index values (not shown) indicate that the Indian monsoon persists but weakens abruptly once elevations are reduced to between 40-60% of modern values, a threshold reported in previous modelling studies (e.g. Abe et al., 2003). As a result of IM weakening, total-column average RH decreases from >90% in CNTL to 70% in TOPO20 and summer precipitation decreases from >6000 mm yr$^{-1}$ in CNTL to 2200 mm yr$^{-1}$ in TOPO20.

ECHAM5 captures a similar threshold behavior in monsoon activity in the EASM region. With lowering of the Himalayan front to 40% of its modern elevations, the broad humid belt that characterizes central China in high elevation scenarios (CNTL, TOPO80, TOPO60, TOPO20a and TOPO20b) shifts southward, resulting in an expanded arid belt in the region (in TOPO40 and TOPO20). This shift in precipitation is associated with a southward retreat of the southwesterly monsoonal winds that penetrate much of eastern Asia (Fig. 3d, e). The southward shift in precipitation and winds is consistent with southward displacement of the EASM documented by the first deposition of aeolian dust in the Miocene (Guo et al., 2008). This observed southward displacement of the EASM is considered to mark the Cenozoic transition of eastern Asian from being dominated by a planetary circulation system to a monsoonal system (e.g. Lu and Guo, 2014; Liu et al., 2017).

### 3.2 Climate response to Himalayan surface-elevation

ECHAM5 experiments TOPO20a and TOPO20b isolate the influence of Himalayan elevations on the regional climate. These experiments generally indicate that the Himalayas, rather than the Tibetan Plateau, govern regional precipitation and circulation patterns. The IM and EASM are strong in both TOPO20a and TOPO20b. The IM is shown by strong low-level wind over the Arabian sea and heavy precipitation across both the Indian subcontinent and the central Himalayas in TOPO20a and TOPO20b (Fig. 3f-g). Likewise, the EASM in these simulations is similar to that in the CNTL as indicated by heavy precipitation to the north of Yangtze river and southerly winds penetrating central China (Fig. 3f-g).

To further elucidate the contribution of the Himalaya to monsoonal dynamics, we calculated the equivalent potential temperature (Fig. S3), which is commonly used to denote the location of monsoonal heating for the Indian Monsoon. In TOPO20a and TOPO20b, the equivalent potential temperature maxima are reduced but in a similar location as in the CNTL case, supporting our conclusion that the Himalayas are the dominant driver of the IM. When the Himalayas are lowered, monsoonal heating decreases and the locus shifts southeastward as cold, dry extratropical air moves southward and mixes



with warm, humid subcontinental air, consistent with the results in Boos (2015), though the extent of the shift is smaller in ECHAM5.

In contrast to this well-established mechanism for the Indian Monsoon, the mechanism for the southward shift of the EASM is not well understood. Uplift of the Himalaya-Tibet orogen and retreat of the Paratethys have been proposed as possible

factors triggering this southward shift between 22-25 Ma (Guo et al., 2008; Liu et al., 2017). The simulation of a strong EASM over central China in both TOPO20a and TOPO20b suggests that uplift of the central and western Himalaya would have been capable of forcing this southward shift.

### 3.3 Model validation of ECHAM5-wiso

Li et al. (2016) demonstrated that ECHAM5 simulates $\delta^{18}O$ variations across the Himalaya and Tibetan Plateau that are in

reasonable agreement with observed precipitation and stream values (see Methods). Consistent with stream water samples, the model captures a decrease in $\delta^{18}O_p$ along a north-south transect across the Himalayas and an increase in $\delta^{18}O_p$ on the Tibetan Plateau (see their Fig. 12d). The main discrepancy occurs on the northwestern Tibetan Plateau in winter and spring. Simulated $\delta^{18}O_p$ is 4 ‰ greater than stream sample values along a cross section extending westward from 85° E and centered on 30° N. Li et al. (2016) attributed this mismatch to local factors, systematic model bias, and the influence of freshwater

discharge from higher altitudes in the watershed.

We further compare ECHAM CNTL $\delta^{18}O_p$ with the modern surface water isotope dataset reported in Li and Garzione (2017) (Fig. S4). Co-located ECHAM and water sample $\delta^{18}O_p$ agree within 2 ‰ at 49.2% of sites, and within 3 ‰ at 73.3% of sites. Several discrepancies between simulated and observed $\delta^{18}O_p$ exist: Firstly, ECHAM $\delta^{18}O_p$ is lower than sampled $\delta^{18}O_p$ over northwestern Tibet (80° E-85° E, 35° N-37° N). This mismatch could be associated with the higher relative humidity in

ECHAM (Fig. S5) than that in observations. This high relative humidity results in weaker evaporation both from land surface and below cloud-base, lowering $\delta^{18}O$ in surface waters. Secondly, ECHAM $\delta^{18}O_p$ is more depleted, by 2-5 ‰, over east-central Tibet (89° E-102° E, 32° N-35° N). The source of this mismatch over east-central Tibet is unclear. An important consideration for both the mismatch over east-central and northern Tibet is that the water samples were collected over a short two-year span and may not reflect mean climatic conditions. This possibility is supported by the large inter-annual variability

in $\delta^{18}O_p$ (up to 9 ‰) in both ECHAM (Li et al., 2016) and in precipitation sample spanning 1986 to 1992 from GNIP (AEA/WMO, 2017).

### 3.4 Moisture source influence on RDM $\delta^{18}O_p$

Under modern elevations, $\delta^{18}O_p$ decreases with elevation on the Himalayan slope, increases with latitude across the Tibetan plateau (Fig. 5a, 6), and varies little (mostly within 2 ‰) on the northern Tibet slope (36° N-40° N) (Fig. 6), consistent with

observations (Li and Garzione, 2017). Under lower elevation scenarios, $\delta^{18}O_p$ values increase relative to the modern



elevation scenario both on the Himalayas and the Tibetan Plateau (Fig. 6). The rate of change of $\delta^{18}O_p$ with elevation and latitude decreases substantially in the monsoonal regions as Tibetan-Himalayan elevations are reduced (Fig. 6a, b).

The oxygen isotope compositions of precipitation on mountain slopes are traditionally assumed to systematically decrease in response to adiabatic cooling, condensation, and rainout of ascending air parcels, a process described by Rayleigh distillation

and the basis for the application of $\delta^{18}O_p$-paleoaltimetry. In this section, we evaluate the degree to which Rayleigh distillation accounts for up-slope decreases in $\delta^{18}O_p$ by comparing RDM to ECHAM $\delta^{18}O_p$ in four separate regions (Fig. 2). Note that the northern Tibet slope is excluded here because most of the northerly air is diverted rather than being forced to ascent (Fig. 3).

For each of the four regions, we initialize the RDM with both fixed moisture sources (with initial T=20°C and RH=80%) as

in paleoaltimetry studies and with ECHAM moisture sources (with T and RH that varies by region and case). The $\delta^{18}O_p$ predicted by the RDM using each moisture source is shown in Fig. 7-8 (red diamonds vs. black circles). With ECHAM-derived T and RH, the RDM $\delta^{18}O_p$ values are close to those simulated by ECHAM5 in all elevation scenarios (Fig. 7-8, Table 1, 3rd column). RDM $\delta^{18}O_p$ with ECHAM-derived moisture is 0-2 ‰ less than that with fixed T and ECHAM RH (Table 1, 4th column), except in the CNTL scenario where the two RDM $\delta^{18}O_p$ are very close. The lower $\delta^{18}O_p$ is due to the

fact that the RH of the initial air parcel is lower in ECHAM than in the prescribed case (Fig. S5). The impact of moisture source differences on $\delta^{18}O_p$ is further decomposed in Table 1 to estimate the contributions adiabatic temperature versus RH changes. As seen in Table 3, the effect of adiabatic temperature changes is consistently small (∼−1 ‰) across all elevation scenarios, reflecting the fact that temperature lapse rates vary little among elevation scenarios (Fig. S1). In contrast, as the initial RH decreases with lowering of elevation, ECHAM-sourced $\delta^{18}O_p$ is lowered by as much as 3.5 ‰ (Table 1, last

column). In sum, these results demonstrate that elevation-related changes in moisture source characteristics substantially impact RDM $\delta^{18}O_p$ estimates.

**3.5 Performance of ECHAM-sourced Rayleigh distillation in the Himalayans**

Our estimates using an RDM implicitly assume that Rayleigh distillation is the dominant process controlling the $\delta^{18}O_p$-elevation relationship. To test this assumption, we compare ECHAM-sourced RDM $\delta^{18}O_p$ and ECHAM5 $\delta^{18}O_p$.

Decreases in $\delta^{18}O_p$ with elevation in ECHAM and the RDM generally agree under high-Himalaya scenarios and are consistent with modern observations (Table 2). Under modern topographic scenarios for the western Himalaya (Fig. S6) and the monsoonal regions (Fig. 7, 8), $R^2$ and p_percent values are greater than 0.77 and 0.51, indicating that RDM and ECHAM $\delta^{18}O_p$ match well. $R^2$ and p_percent values are similarly high, above 0.68 and 0.53, for the western Himalaya and the monsoonal regions in high-Himalaya scenarios (TOPO80, TOPO60, TOPO20a, and TOPO20b in the IM region; TOPO80,

TOPO20b in the EASM region) again indicating a good match between ECHAM and RDM $\delta^{18}O_p$ and suggesting that Rayleigh distillation drives isotopic compositions in these regions.

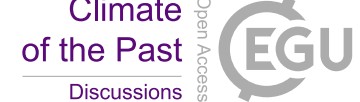



In other regions and under low-elevation scenarios, however, the comparison between RDM and ECHAM $\delta^{18}O_p$ is poor (Table 2). For instance, on the western Himalayas, in the TOPO80 case (Fig. S6), the lapse rate of ECHAM $\delta^{18}O_p$ is much smaller than the lapse rate predicted by the RDM (Fig. S6), as shown by p_percent values of less than 0.15. Under even lower elevation scenarios (TOPO60, TOPO40, TOPO20), orographic precipitation is not triggered over the Himalayas (Fig.

S6c-e), making the RDM an unsuitable representation of precipitation processes. In the transitional region, the $\delta^{18}O_p$ can vary by more than 5‰ at a specific elevation under all elevation scenarios (Fig. S7). This large spread is represented by a low $R^2$ value of 0.12 for the CNTL case, and even lower values (less than 0.10) for other topographic scenarios. In TOPO40 and TOPO20, ECHAM $\delta^{18}O_p$ shows little relationship with elevation (Fig. 6d).

In the monsoonal regions, the relationship between ECHAM5 $\delta^{18}O_p$ and elevation is weak in low elevation scenarios (Table

2 and Fig. 6) and compares poorly with the RDM. In the IM region, p_percent values for the TOPO40 and TOPO20 are less than 0.29. In the EASM region, ECHAM5 $\delta^{18}O_p$ is higher than RDM $\delta^{18}O_p$ (Fig. 7, 8) in the TOPO60 scenario and the agreement is low (with p_percent value of 0.31). Under even lower elevation scenarios (TOPO40 and TOPO20), ECHAM $\delta^{18}O_p$ shows no relationship with elevation (Table 2 and Fig. 6b) and also compares poorly with the RDM (with p_percent less than 0.29).

Among the high-Himalaya-low-Tibet cases (TOPO20a and TOPO20b), RDM and ECHAM $\delta^{18}O_p$ match well in the IM region. The match is similarly good for TOPO20b in the EASM region. However, in the case with a low eastern flank (TOPO20a), the comparison is poor with an $R^2$ of 0.006 and a p_percent of −0.05.

### 3.6 Factors influencing $\delta^{18}O_P$-elevation relationship on the Himalayan slope

As shown in 3.5, Rayleigh distillation cannot explain $\delta^{18}O_p$ variations with elevation for most regions in the reduced

elevation (TOPO40, TOPO20) scenarios and in many regions under higher elevation scenarios (TOPO80, TOPO60). To understand the factors influencing $\delta^{18}O_p$, we quantify the mass fluxes of $^{18}O$ (Fig. 9), distinguishing the processes that increase the mass flux of $^{18}O$ of the column (mixing and surface recycling) from those that decrease it (Rayleigh distillation and convective rainfall). Note that this method of taking the vertical column as a whole does not isolate the processes occurring within this air column (e.g. sub-cloud re-evaporation) but this limitation does not impact the ability to identify the

contribution of Rayleigh distillation and local processes. The results from this method yield very different contributions on the western slope from those in other regions, thus, the western Himalayas are reported separately.

On the western Himalayas, local surface recycling and convective rainfall contributes substantially to the total mass flux of $^{18}O_p$ in the highest elevation scenarios (Fig. 9a). The contributions from these processes accounts for the poor match between RDM and ECHAM $\delta^{18}O_p$ in this region. In the TOPO20a and TOPO20b scenarios, enhanced transport of enriched vapor

from the south (see Fig. S8, 1 of 42 trajectories in CNTL versus 11 and 13 of 42 in TOPO20a and TOPO20b) increases the contribution from vapor mixing.



In the monsoonal regions and the transitional region, vapor mixing is the predominant source of $^{18}$O under high-elevation scenarios and reflects the advection of $\delta^{18}$O-enriched vapor from the Arabian Sea and Bay of Bengal. This mass flux of vapor mixing decreases substantially with a reduction in elevation, whereas the mass flux of surface recycling remains approximately constant in all elevation scenarios. As a result, surface recycling is as or more important than mixing as a source of $^{18}$O under low-elevation scenarios (Fig. 9c, d). This change in relative importance of the two sources represents an increase in the importance of local versus remote sources as monsoon strength weakens under low-elevation scenarios.

In high elevation scenarios, Rayleigh distillation acts as the dominant sink of $^{18}$O in the monsoonal regions causing $\delta^{18}$O to decrease markedly with elevation (Fig. 9c-d). Under reduced elevation scenarios, the absolute mass flux from Rayleigh distillation decreases, as large-scale precipitation due to stable upslope ascent decreases and convective precipitation increases (Fig. 10). With reduced elevation, the percentage of large-scale precipitation to total precipitation falls from 86% to 18% in the transitional region, from 93% to 18% in the IM region and from 80% to 15% in the EASM region, mirroring the decrease in the mass contribution of Rayleigh distillation, which falls from 85% to 11% in the transitional region, from 93% to 22% in the IM region and from 80% to 18% in the EASM region. As a result of the reduction in precipitation by stable upslope ascent, convective precipitation is the largest $^{18}$O sink in TOPO20 and TOPO40 in both the transitional region and the IM region, and in TOPO60, TOPO40, TOPO20 and TOPO20a in the EASM region. These are also the scenarios that exhibited a poor match between RDM $\delta^{18}$O$_p$ and ECHAM $\delta^{18}$O$_p$ (Sect. 3.5). A similar reduction in large-scale precipitation with reduction in elevation is also captured in the Andes region (Insel et al., 2009) and is associated with a decrease in the rate of change of $\delta^{18}$O$_p$ with elevation.

Note that, although Rayleigh distillation is the primary sink under high-elevation scenarios, RDM $\delta^{18}$O$_p$ does not match ECHAM $\delta^{18}$O$_p$ well in the transitional region because of the large spread in ECHAM $\delta^{18}$O$_p$ (Fig. S7). This spread is due to the bifurcated sources from northwest India (relatively enriched) and the Bay of Bengal (relatively depleted), since air parcels follow separate trajectories before mixing at the peak (Fig. S8).

In sum, the mismatch between RDM and ECHAM $\delta^{18}$O$_p$ is caused by a weakening of Rayleigh distillation under low-elevation scenarios, triggered by a reduction in large-scale precipitation.

### 3.7 $\delta^{18}$O$_p$-latitude relationship on the Tibetan Plateau

In contrast to $\delta^{18}$O$_p$ on the Himalayas, $\delta^{18}$O$_p$ on the Tibetan Plateau increases linearly with latitude. It has been proposed that past $\delta^{18}$O$_p$, reconstructed from isotopic analyses of ancient soil and lake carbonates, could provide information about past elevations on the Tibetan Plateau after removing the modern meridional $\delta^{18}$O$_p$ gradient (Bershaw et al., 2012). This proposal is problematic because the meridional $\delta^{18}$O$_p$ gradient and the processes that set this gradient on the Tibetan Plateau are poorly understood today and were likely different in the past in ways that are not known.

Tibetan Plateau meridional $\delta^{18}$O$_p$ gradients simulated by ECHAM5 under modern and reduced elevation scenarios are shown in Fig. 11. The average meridional gradient in CNTL is 0.95 ‰/°, very close to the observed gradient of 1.09 ‰/° (Li and



Garzione, 2017). Three features of the latitudinal $\delta^{18}O_p$ gradients are notable. Firstly, the gradient varies with longitude (Fig.11a, b) and variations are larger in high-elevation cases (e.g. from 1.90 ‰/° to 0.20 ‰/° in CNTL). Secondly, a linear fit is generally good with a high coefficient of determination ($R^2 > 0.8$) east of 85°W for all cases but TOPO20a and TOPO20b (Fig. S9). The high goodness of fit indicates a robust $\delta^{18}O_p$-latitude relationship for the cases with uniform reductions in topography. The poor fit in TOPO20a and TOPO20b is due to the larger variations in elevation across the Tibetan Plateau. In these two cases, dry conditions (Fig. S5) on the steep leeward side of the Himalaya favor strong below cloud-base re-evaporation, increasing local latitudinal $\delta^{18}O_p$ gradients. Lastly, in cases with uniform lowering of topography, the median meridional gradient (Fig. 11b) decreases almost linearly with reductions from 100% to 60% of modern topography, changes more abruptly between 60% and 40%, and varies little once the topography is lowered below the monsoon threshold in TOPO40 and TOPO20.

The modern latitudinal $\delta^{18}O_p$ gradient on the Tibetan Plateau has been attributed to both surface recycling and sub-cloud re-evaporation. Surface recycling increases $\delta^{18}O_p$ by adding more enriched vapor from the land surface, while sub-cloud evaporation enriches $^{18}O$ in precipitation by partial evaporation of falling raindrops in an unsaturated air column. To understand the cause of the decline in the meridional $\delta^{18}O_p$ gradient, we quantified both the sources of $^{18}O$ (surface recycling and vapor mixing) and the isotopic contribution due to sub-cloud re-evaporation. Sub-cloud evaporation is quantified separately because this effect is contained in but cannot be isolated from the sources of $^{18}O$.

Surface recycling is the primary source of $^{18}O$ on the Tibetan Plateau with largely consistent contributions under all elevation scenarios (Fig. 12). The mass flux of $^{18}O$ due to vapor mixing is small and becomes a sink in low-elevation scenarios, since there are increasingly more depleted sources from the south in these scenarios (e.g., 17 out of 42 in CNTL vs. 30 out of 42 in TOPO20 for one location (33° N, 90° E)). To further quantify the contribution of surface recycling to total precipitation, we subtracted $\delta^{18}O$ in precipitation from $\delta^{18}O$ in recycled vapor and normalized by the amount of precipitation and recycled vapor as in the equation:

$$\left(\delta^{18}O_s - \delta^{18}O_p\right) \times E/P \times \rho_{water}, \tag{7}$$

where $\delta^{18}O_p$ is the isotopic composition in precipitation; $\delta^{18}O_s$ is the condensate of recycled vapor; $E$ (m h$^{-1}$) is the surface evaporation rate and $P$ (m h$^{-1}$) is the total precipitation rate. Results from Eq. (7) show that $\delta^{18}O$ of surface recycling contributes less than 0.1 ‰ in CNTL and slightly more (−2.67 ‰) in TOPO20 (Table 3). The overall small contribution from surface recycling in ECHAM5 is due to the fact that $\delta^{18}O$ values in soil are similar to those in precipitation (Fig. S10). This small contribution grows in low-elevation scenarios due to the increased fraction of evaporated vapor to total precipitation (Fig. S11). Nonetheless, surface recycling plays a secondary role in decreasing the meridional $\delta^{18}O_p$ gradients.

Sub-cloud re-evaporation occurs within an unsaturated air column as falling raindrops undergo kinetic fractionation and become isotopically enriched (Stewart, 1975). Enrichment is reduced in heavier rain where the relative humidity is high, resulting in the observed anti-correlation between $\delta^{18}O_p$ and precipitation rate referred to as the amount effect. To quantify this enrichment due to sub-cloud re-evaporation, we show $\delta^{18}O_p$ against daily precipitation (Fig. 13). The slope of this $\delta^{18}O_p$-



precipitation relationship denotes the strength of the amount effect. In low-elevation scenarios, the $\delta^{18}O_p$-precipitation slope is shallower (Fig. 13), indicating a weaker amount effect and stronger sub-cloud re-evaporation even at high precipitation rates.

This shallow slope of sub-cloud evaporation results in the shallower meridional $\delta^{18}O_p$ gradient on the Tibetan Plateau in low-elevation scenarios. In each individual elevation scenario, the enrichment of $\delta^{18}O_p$ due to sub-cloud evaporation is stronger at higher latitudes since there are more events of lower precipitation rates at higher latitudes than at lower latitudes (Fig. 14). As a result of this different distribution of precipitation, rainfall at higher latitudes is more enriched than that at lower latitudes. To further compare this contribution of sub-cloud re-evaporation with surface recycling, the excess enrichment of $\delta^{18}O_p$ due to sub-cloud re-evaporation is quantified by the $\delta^{18}O_p$ difference at high and low precipitation rates, weighted by

precipitation rates (Table 3). The excess enrichment due to sub-cloud re-evaporation is much larger than that due to surface recycling and decreases with reduced elevation.

To explain this stronger sub-cloud re-evaporation in low-elevation scenarios, we refer to the kinetic fractionation process in ECHAM5 during partial evaporation of raindrops. As shown in Hoffmann et al. (1998), kinetic fractionation in ECHAM5 is formulated as:

$$\alpha = \frac{RH}{\frac{D}{\widehat{D}}(RH-1)+1},$$                                                                         (8)

where $\alpha$ is the fractionation factor; $D \times \widehat{D}^{-1}$ represents the ratio of diffusivities between $^{16}O$ and $^{18}O$ and has the constant value of 0.9727; $RH$ is the effective relative humidity of the grid box. To estimate how this kinetic fractionation changes between different elevation scenarios, we approximated the effective relative humidity to be the total-column-averaged RH. As seen in Fig. S12, the total-column-averaged RH at any given precipitation rate is higher in high-elevation scenarios than

that in low-elevation scenarios. Specifically, when the precipitation rate is very high at 40mm day$^{-1}$, the RH is at ~100% in CNTL, suggesting very little kinetic fractionation and weak sub-cloud re-evaporation. In comparison, in TOPO20 the RH is much lower at ~85%, indicating the presence of sub-cloud re-evaporation even at high precipitation rates.

In sum, meridional $\delta^{18}O_p$ gradients on the Tibetan Plateau decrease with lower elevation, and this reduction is due to stronger sub-cloud evaporation in low-elevation scenarios.

**4 Discussion**

**4.1 Processes impacting paleoaltimetry**

The use of $\delta^{18}O_p$ as a paleoaltimeter is based on the principle that atmospheric vapor becomes saturated and condenses under forced ascent in orographic regions leading to preferential rainout of $^{18}O$. Observations of meteoric $\delta^{18}O$ in orographic regions support the use of stable isotope paleoaltimetry and show a robust and significant decreasing relationship in $\delta^{18}O_p$

with surface elevation (e.g. Poage and Chamberlain, 2001; Fiorella et al., 2015; Li and Garzione, 2017) that can be modeled



by Rayleigh distillation (Rowley and Garzione, 2007). However, there is no a priori reason that modern $\delta^{18}$O-elevation relationships should hold in the past when surface elevations and associated atmospheric conditions were different (e.g. Ehlers and Poulsen, 2009; Poulsen et al., 2010; Feng et al., 2013; Botsyun et al., 2016).

Our ECHAM5 results confirm that the processes that govern $\delta^{18}O_p$ vary spatially and change in response to changes in 5 surface elevation. Across the Himalayan front, $\delta^{18}O_p$ generally decreases with elevation, particularly in scenarios with high Himalayan elevations. However, Rayleigh distillation often does a poor job of explaining the $\delta^{18}O_p$-elevation relationship in the Himalaya-Tibetan region. There are two primary reasons. Firstly, local processes including convection and surface recycling can dominate the land-surface exchange of vapor (Fig. 9). These local processes are especially strong under low-elevation scenarios in monsoonal regions, where forced ascent is weak, and in the western Himalayas. Secondly, in regions 10 with multiple moisture sources, mixing of air masses with different $\delta^{18}$O can cause the $\delta^{18}O_p$-elevation signal to deviate from that expected from Rayleigh distillation. The transitional region on the Himalaya, which receives moisture from both the Arabian Sea and the Bay of Bengal, is such a case (Fig. S8). On the Tibetan Plateau, the meridional $\delta^{18}O_p$ gradient decreases in response to reduced elevation. This meridional $\delta^{18}O_p$ gradient is primarily controlled by sub-cloud re-evaporation under modern elevations. Sub-cloud re-evaporation increases due to wetter conditions in low-elevation scenarios, reducing the 15 meridional $\delta^{18}O_p$ gradient. Our conclusions are similar to those of Feng et al. (2013) for the North American Cordillera, in which it was shown that the isotopic fractionation of precipitation was not primarily due to Rayleigh distillation.

Our results also highlight the large influence that the choice of moisture source characteristics has on RDM $\delta^{18}O_p$ and are consistent with those of Botsyun et al. (2016). When we use fixed T and RH, ECHAM5 and RDM $\delta^{18}O_p$ agreement is poor with considerable enrichment in RDM. When we use ECHAM5 T and RH, the agreement is considerably improved. This 20 change in moisture sources represents elevation-induced climate change unrelated to rainout and not captured by Rayleigh distillation. The overall impact from temperature and RH (red diamonds in Fig. 7-8) is a slight underestimation in high-elevation scenarios and severe overestimation in low-elevation scenarios (e.g. by ~100% in TOPO20 in the IM region).

### 4.2 Implications for $\delta^{18}O_p$ paleoaltimetry

As discussed above, $\delta^{18}O_p$ paleoaltimetry is only appropriate for monsoonal regions in high elevation scenarios. Nonetheless, 25 proxy $\delta^{18}O_p$ from sites across the Himalaya-Tibet region have been used to infer paleoaltimetry throughout the Cenozoic (Fig. 15, Table S1). We classify these sites into five types indicating their utility for $\delta^{18}O_p$ paleoaltimetry.

As shown in Fig. 15 (black dots), the modern $\delta^{18}O_p$-elevation relationship is well captured by ECHAM5 and well represented by the Rayleigh distillation process under high-elevation scenarios for 7 of 50 sites located on the high Himalayas. However, for these seven sites, the $\delta^{18}O_p$-elevation relationship breaks down as height of the Himalaya-Tibet 30 orogen is reduced. For 13 of 50 sites in central Tibet (green dots), there is no direct $\delta^{18}O_p$-elevation relationship, since elevation is largely uniform and $\delta^{18}O_p$ increases linearly with latitude due to sub-cloud re-evaporation and recycling (as shown by 3.7). Although it has been proposed that $\delta^{18}O_p$-paleoaltimetry is still possible after removing this meridional $\delta^{18}O_p$



gradient from $\delta^{18}O_p$ (Bershaw et al., 2012), the meridional gradient varies by as much as 70% with elevation and is largely unknown in the past (as shown by 3.7); thus, $\delta^{18}O_p$-paleoaltimetry is not appropriate for these sites. For 7 of 50 sites (red dots), elevations are too low so that $\delta^{18}O_p$ has no relationship with elevation and changes very little throughout different elevation scenarios (Fig. 6). As a result, $\delta^{18}O_p$ is not indicative of elevation. For sites in the western and transitional regions

(6 of 50), $\delta^{18}O_p$ either relates very little to elevation or exhibits a large range at the same elevation (Fig. S7), and is thus not appropriate for $\delta^{18}O_p$-paleoaltimetry. Lastly, at sites in northern Tibet (yellow dots, 15 of 50), moisture is mostly diverted by the Tibetan Plateau, Rayleigh distillation is not triggered, and $\delta^{18}O_p$ does not vary with elevation (Fig. 6).

Although $\delta^{18}O_p$-elevation relationship at sites in monsoonal regions (Fig. 15, black dots) are well explained by Rayleigh distillation in our experiments, $\delta^{18}O_p$-paleoaltimetry should still be applied with care for these sites even under high

elevation conditions due to the influence of climate change. Proxy $\delta^{18}O_p$ values as low as −18-−16 ‰ are reported for the early Eocene (Ding et al., 2014) and around −14 ‰ for the late Eocene (Rowley and Currie, 2006). These values almost certainly reflect high elevations. Nonetheless, elevation-independent factors, including atmospheric $pCO_2$ (Poulsen and Jeffrey, 2011) and paleogeography (Roe et al., 2016), add substantial uncertainty to the quantification of past surface elevations.

**4.3 Caveats**

Like all models, ECHAM has limitations. Most pertinent to this study, ECHAM simulates higher precipitation along the steepest slope of large mountain than indicated by satellite observations (Roe et al., 2016 and reference there in), which is a common problem in most GCMs. The model tends to overestimate the modern precipitation amount and RH on the western Tibetan plateau. This overestimation also exist in earlier version of ECHAM (Roe et al., 2016) and other models such as the

LMDZ-iso (Zhang and Li, 2016). The overestimation of total-column-averaged RH might weaken sub-cloud re-evaporation process on the western plateau, and this weaker re-evaporation could potentially lower the meridional $\delta^{18}O_p$ gradient in this region. Another limitation of the model is that sub-grid scale lakes are not included. Although it has been proposed that the lakes provide $\delta^{18}O$-enriched vapor to the air (Bershaw et al., 2012), under equilibrium conditions the net $\delta^{18}O$ flux to the air should be zero. Despite these limitations, ECHAM's simulation of $\delta^{18}O_p$ compares favorably with natural water isotopic

measurements (Li et al., 2016).

In this study, we use a series of idealized simulations to investigate the response of water isotopes to mountain uplift and to understand the mechanisms that control $\delta^{18}O_p$ variations on the Himalayas and the Tibetan Plateau. We do not compare and evaluate our model results directly against proxies, since the simulations are not meant to represent specific time slices in the geologic past, and do not include changes in time-specific boundary conditions such as greenhouse gas composition,

vegetation, orbital parameters, glacial boundary conditions, tectonic displacement of the Indian subcontinent, or inclusion of the Parative. Changes in these boundary conditions, consistent with those that occurred through the Cenozoic, are likely to have a substantially influence on simulated $\delta^{18}O_p$. For instance, high early Cenozoic atmospheric $CO_2$ may have increased





$\delta^{18}O_p$ over high elevation regions (Jeffery et al., 2012; Poulsen and Jeffery, 2011) and on the Tibetan Plateau by as much as 8‰ (Poulsen and Jeffery, 2011). In addition, Earth's orbital variations have been shown to contribute as much as 7 ‰ to oxygen isotope changes on the Tibetan Plateau (Battisti et al., 2014), which is comparable to the isotope difference from CNTL to TOPO20 (about −11 ‰) in Fig. 6. These large variations associated with orbital fluctuations are not observed in

long records spanning $10^6$ yrs (Deng and Ding, 2015; Kent-Corson et al., 2009), presumably because the terrestrial proxy archives of $\delta^{18}O$ integrate over orbital time periods.

## 5 Conclusion

The isotopic composition of ancient meteoric waters archived in terrestrial proxies is often used as a paleoaltimeter under the assumption that rainout during stable air parcel ascent over topography leads to a systemic isotopic depletion through

Rayleigh distillation. We use an isotope-enabled GCM, ECHAM5-wiso, to evaluate the extent to which oxygen isotopes can be used as a paleoaltimeter for the Himalaya-Tibet region and to explore the processes that control the $\delta^{18}O$-elevation relationship. Overall, our study highlights the myriad processes that influence $\delta^{18}O_p$ in the Himalaya-Tibet region now and during its uplift.

We find that Rayleigh distillation describes most of the $\delta^{18}O_p$ variation with elevation in the monsoonal regions under high-

topography scenarios. In contrast, Rayleigh distillation does a poor job of describing $\delta^{18}O_p$ variation with elevation under high-topography scenarios in the western Himalayas due to the dominance of local convection and surface recycling in that region. When the Himalaya-Tibet elevations are reduced to below one-half of their modern heights, $\delta^{18}O_p$ exhibits no relationship with elevation. At these reduced elevations, $\delta^{18}O$ fractionation occurs primarily through local convection and surface recycling. On the Tibetan Plateau under modern elevations conditions, $\delta^{18}O_p$ linearly increases with latitude

primarily due to sub-cloud re-evaporation. The $\delta^{18}O_p$ gradient decreases as the plateau is lowered due primarily to stronger sub-cloud re-evaporation under drier conditions and secondarily to increased moisture sources from surface recycling. Because of these elevation-independent processes, we conclude that only 7 out of the 50 paleoaltimetry sites are appropriate for $\delta^{18}O$-paleoaltimetry. Taken together, these results indicate that stable isotope paleoaltimetry in the Himalaya-Tibet region, as in other orogenic regions, is at best a blunt instrument for inferring past surface elevations.

**Author contribution**

C. Poulsen designed and ran the simulations. H. Shen performed the model analyses. Both authors contributed to interpreting the results and writing the manuscript.



**Acknowledgements**

This work was supported by National Science Foundation Grant F033233 to C. Poulsen.

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





**Table 1. Contribution of physical processes to $\delta^{18}O_p$ (all units in ‰) averaged over monsoonal regions (IM and EASM). In the table, $RDM_{ECHAM}$ represents RDM initiated using the moisture sources (i.e. air temperature and relative humidity) from ECHAM5; $RDM_{Fixed}$ is the $\delta^{18}O_p$ simulated by RDM initiated with fixed moisture source of T=20°C and RH=80%; and**

5     **$RDM_{Fixed\_T}$ is initiated with fixed T=20°C and ECHAM RH (refer to section 2.2 for more details on the three different moisture sources). All columns show values averaged from mountain foot to mountain peak, except for the 2nd column showing the difference in $\delta^{18}O_p$ between mountain peak and foot as simulated by $RDM_{ECHAM}$.**

| Case | $RDM_{ECHAM}$ peak – foot ( ‰) | $RDM_{ECHAM}$ – ECHAM5 ( ‰) | $RDM_{Fixed}$ – $RDM_{ECHAM}$ ( ‰) | $RDM_{Fixed\_T}$ – $RDM_{ECHAM}$ ( ‰) | $RDM_{Fixed}$ – $RDM_{Fixed\_T}$ ( ‰) |
|---|---|---|---|---|---|
| CNTL | -8.00 | 0.88 | -0.32 | -1.52 | 1.20 |
| TOPO80 | -6.15 | 1.89 | 1.73 | -0.96 | 2.70 |
| TOPO60 | -4.16 | 1.80 | 2.25 | -1.00 | 3.25 |
| TOPO40 | -2.53 | 0.86 | 2.34 | -1.23 | 3.57 |
| TOPO20 | -1.12 | -0.30 | 1.86 | -1.51 | 3.37 |



**Table 2. Summary of the comparison between ECHAM and RDM $\delta^{18}O_p$ lapse rates under different topographic scenarios in different climate regions. Regions and scenarios with high $R^2$ and p_percent are marked with an "X" indicating the existence of a significant $\delta^{18}O_p$-elevation relationship (see methods in section 2.2 for definitions of $R^2$ and p_percent).**

| Region | CNTL | TOPO80 | TOPO60 | TOPO40 | TOPO20 | TOPO20a | TOPO20b |
|--------|------|--------|--------|--------|--------|---------|---------|
| Western Transitional | X | | | | | X | X |
| IM | X | X | X | | | X | X |
| EASM | X | X | | | | | X |



**Table 3. The isotopic contribution (in ‰) due to sub-cloud re-evaporation and surface recycling on the Tibetan Plateau for different elevation scenarios.**

|  | CNTL | TOPO80 | TOPO60 | TOPO40 | TOPO20 |
|---|---|---|---|---|---|
| Sub-cloud re-evaporation | 9.06 | 10.02 | 7.97 | 5.21 | 5.28 |
| Surface recycling | 0.09 | 0.14 | -1.40 | -2.67 | -1.8 |





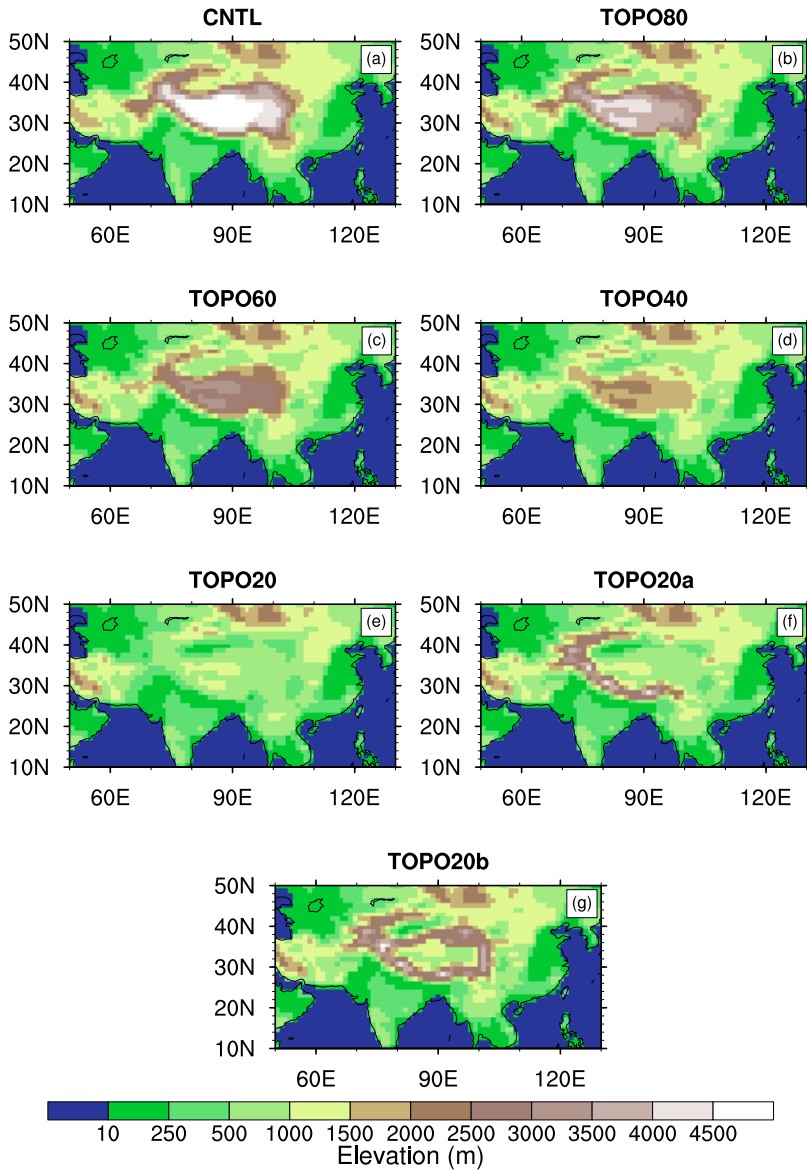

**Figure 1.** Surface elevations (m) prescribed in our ECHAM5 cases (a) CNTL, (b) TOPO80, (c) TOPO60, (d) TOPO40, (e) TOPO20, (f) TOPO20a, and (g) TOPO20b. Note that the CNTL simulations includes modern elevations. The names of the other cases (e.g. TOPO60) indicate surface elevations of the Himalaya-Tibetan region relative to the modern (e.g. 60%). Cases TOPO2a and TOPO20b are modifications of the TOPO20 case (see Methods).



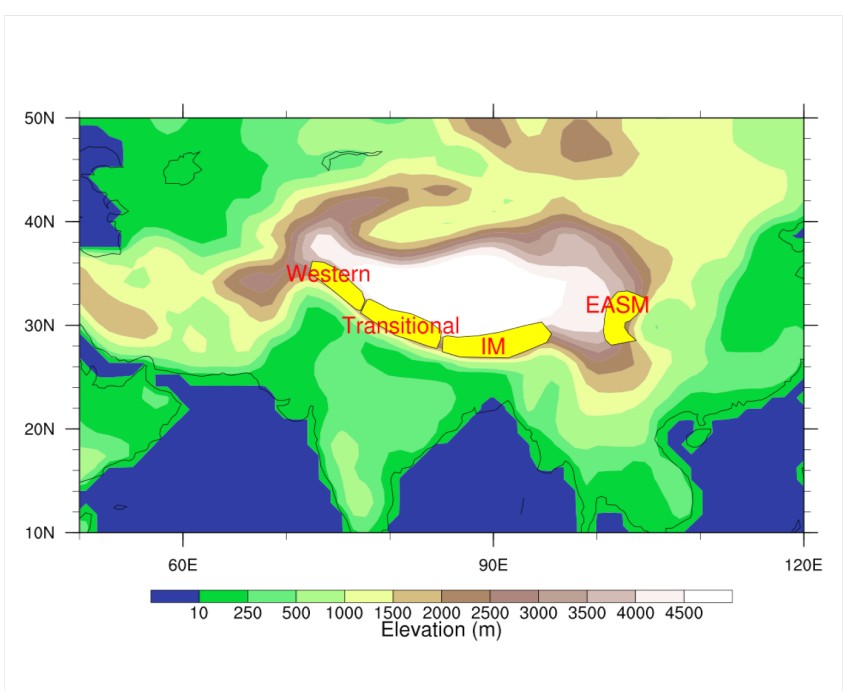

5 **Figure 2. Topographic map (m) showing Himalayan climate zones after Yao et al. (2013). Western, transitional, Indian Summer Monsoon (IM), and Eastern Asian Summer Monsoon (EASM) regions are marked. Transitional areas between central and eastern Himalaya are excluded because climate and isotopic signals are similar to IM and EASM regions.**





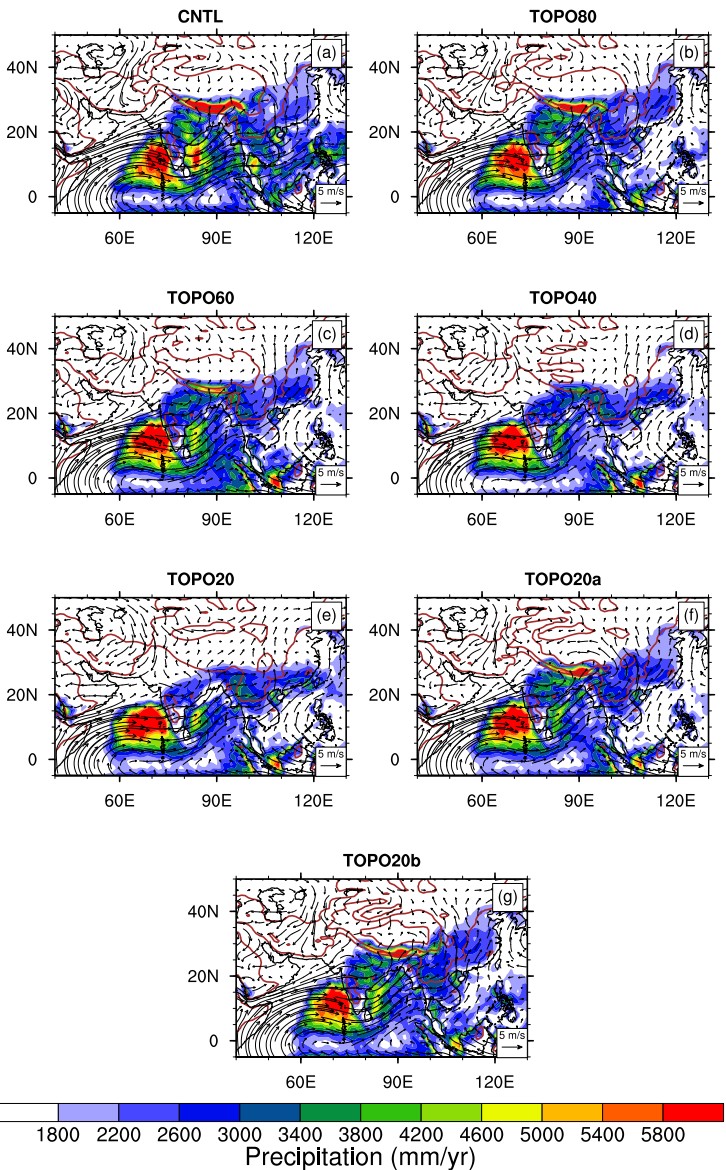

**Figure 3. Summer (June-July-August) low-level (850 hPa) wind (arrows) and precipitation (shaded) for (a) CNTL (b) TOPO80, (c) TOPO60, (d) TOPO40, (e) TOPO20, (f) TOPO20a, and (g) TOPO20b cases. Brown contour lines represent 500 m and 2000 m surface elevation contours in the Tibetan region.**



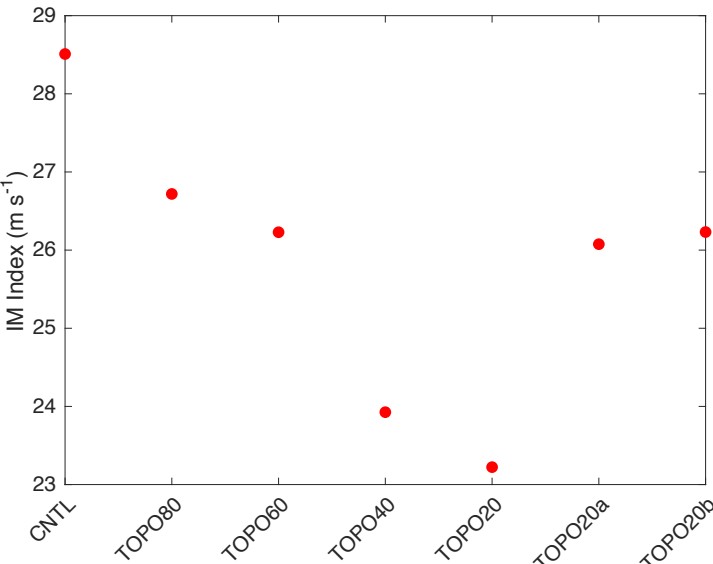

**Figure 4. Indian Summer Monsoon (June-July-August) index, WSI1, calculated as the vertical wind shear between the lower (850 hPa) and upper (200 hPa) troposphere in the region (5-20° N, 40-80° E).**



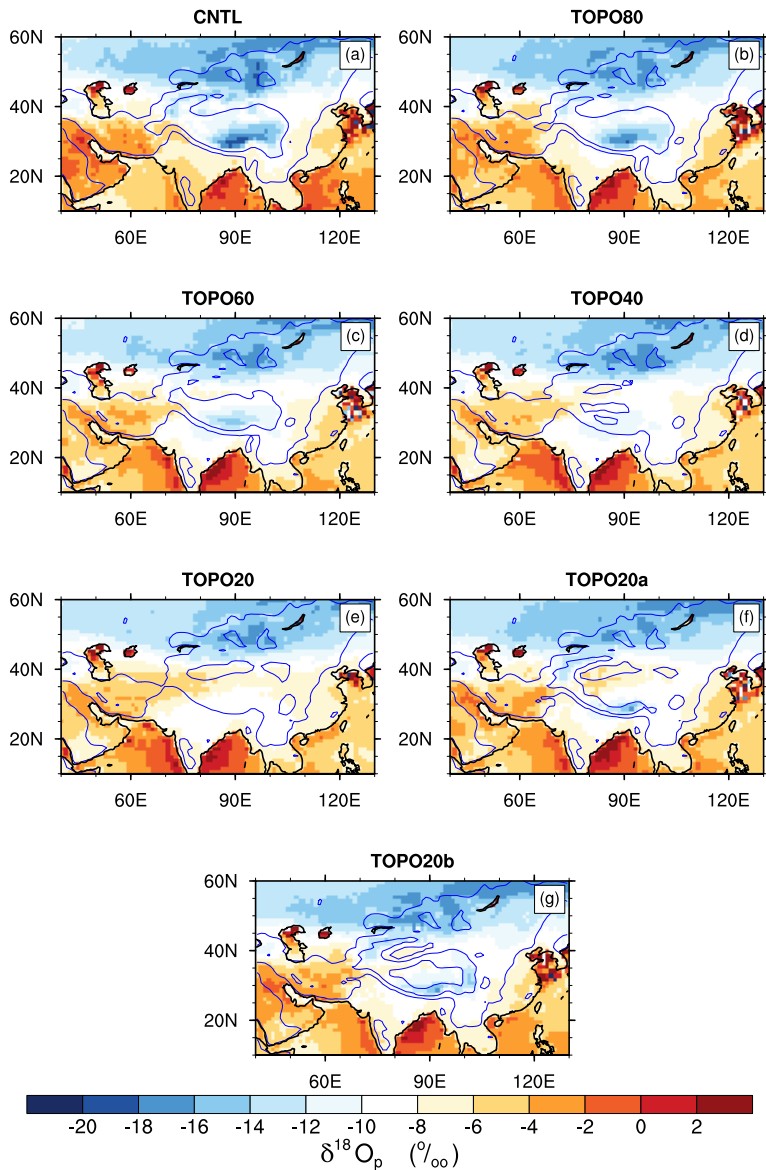

**Figure 5. Monthly-precipitation-weighted ECHAM δ¹⁸Oₚ (in ‰) for cases (a) CNTL, (b) TOPO80, (c) TOPO60, (d) TOPO40, (e) TOPO20, (f) TOPO20a, (g) TOPO20b. Light blue contour lines represent 500 and 2000 m surface elevation contours.**





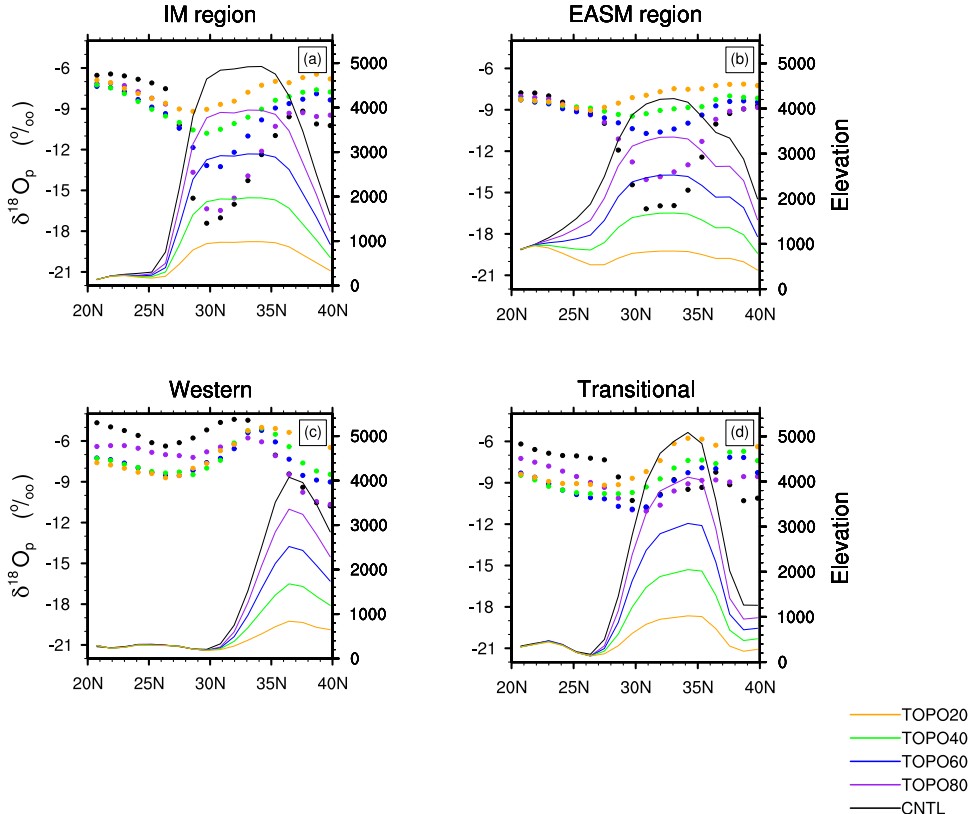

**Figure 6. Summer δ¹⁸Oₚ (filled circles) and surface elevations (lines) along latitudinal transects for the (a) IM; (b) EASM; (c) western; and (d) transitional regions. These values represent zonal-average values over the specific region.**



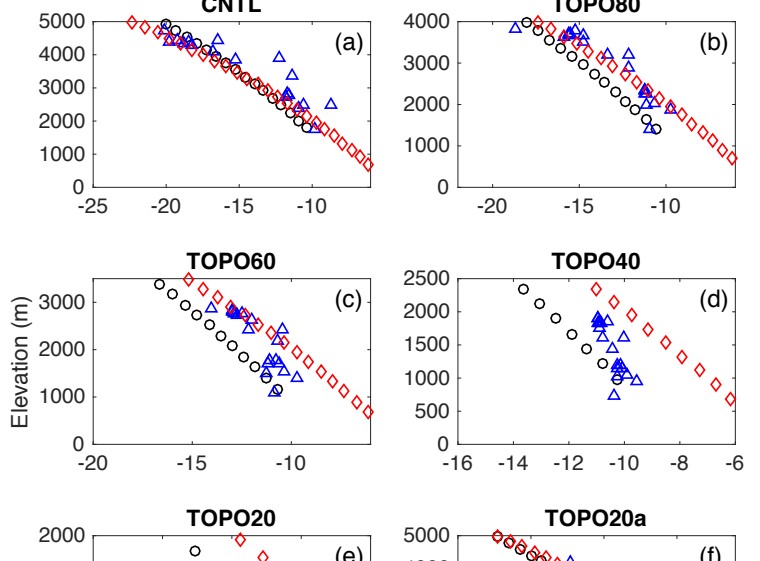

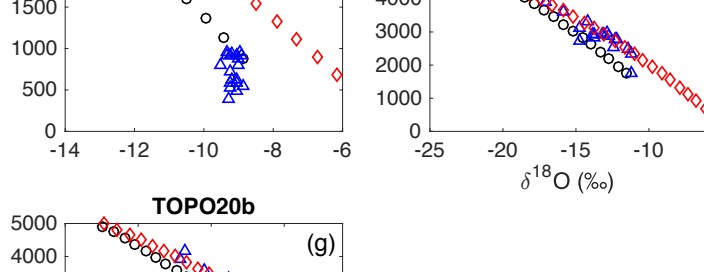

**Figure 7. Monthly-precipitation-weighted $\delta^{18}O_p$ (‰) versus elevation (m) for the Indian Summer Monsoon (ISM) region of the southern Himalayan flank as simulated by ECHAM5 (blue triangle), RDM initiated with ECHAM5 moisture sources (black circle) and RDM initiated with fixed moisture sources of T=20°C and RH=80% (red diamond) for the (a) CNTL, (b) TOPO80, (c) TOPO60, (d) TOPO40, (e) TOPO20, (f) TOPO20a, and (g) TOPO20b cases.**





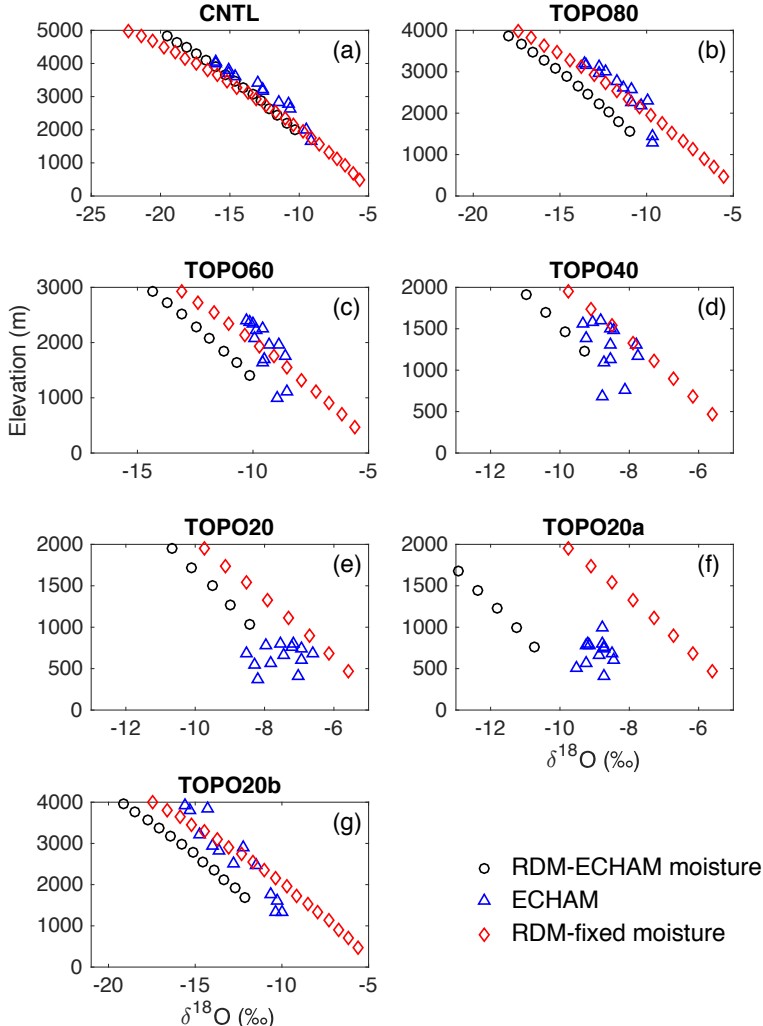

Figure 8. Same as in Figure 7 but for the East Asian Summer Monsoon (EASM) region of the southern Himalayan flank.





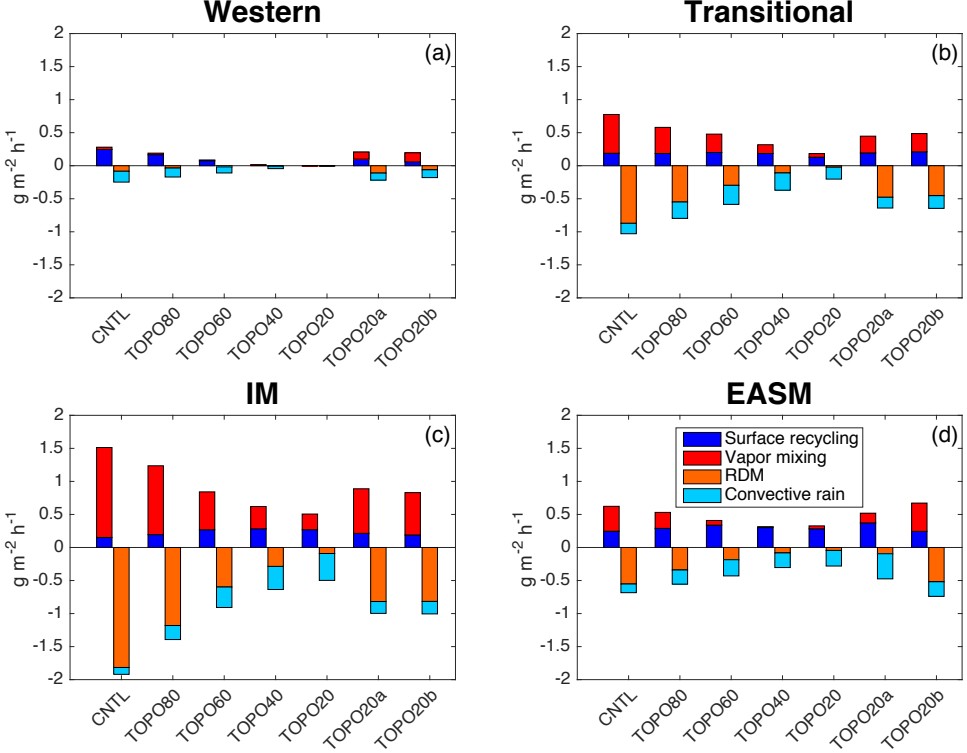

**Figure 9.** Mass flux of $^{18}$O (g m$^{-2}$ h$^{-1}$) for the (a) western Himalayas, (b) the transitional, (c) IM and (d) the EASM regions. Sources (positive values) and sinks (negative values) balance within 20% or 0.1 g m$^{-2}$ h$^{-1}$ for fluxes that are close to zero. (Small errors in the net balance arise due to the centered finite-difference method used to calculate derivatives in the advection terms.) Note that

5 the largest source and sink varies by region and with elevation. Local convection and surface recycling are the dominant source and sink of $^{18}$O in the western Himalayas and under low-elevation scenarios, while Rayleigh distillation and vapor mixing dominate in high-elevation scenarios in the monsoonal and transitional regions.



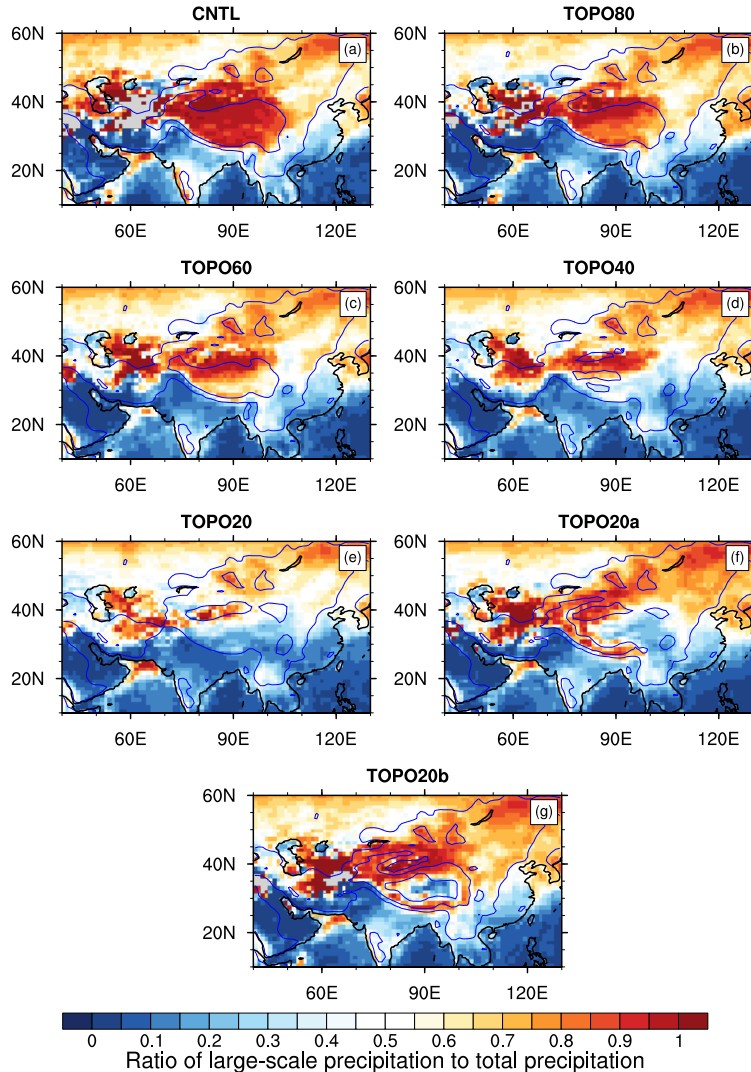

**Figure 10.** Ratio of large-scale to total precipitation rate in the (a) CNTL, (b) TOPO80, (c) TOPO60, (d) TOPO40, (e) TOPO20, (f)
TOPO20a, and (g) TOPO20b cases. Light blue lines mark the 500 and 2000 m elevation contours in each case. In the Himalayas,
this ratio decreases when mountain elevations are reduced. As a result, large-scale precipitation is dominant in high-elevation
scenarios, but not in low-elevation scenarios.



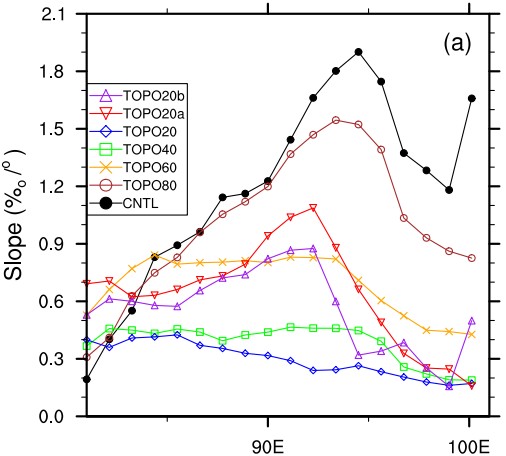

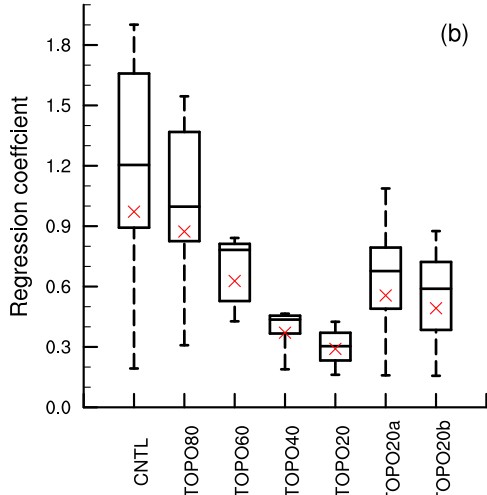

**Figure 11. (a) The slope (in ‰/°) of linear regression of $\delta^{18}O_p$ on latitude from ECHAM5 output, used as an approximation for the meridional gradient of $\delta^{18}O_p$. (b) Boxplot shows the variability of the meridional gradient of $\delta^{18}O_p$. Red "x" indicates mean values for each scenario, calculated by regressing longitudinally averaged $\delta^{18}O_p$ across the Tibetan Plateau on latitude. Box plots show minimum, maximum, median, and quartile values.**





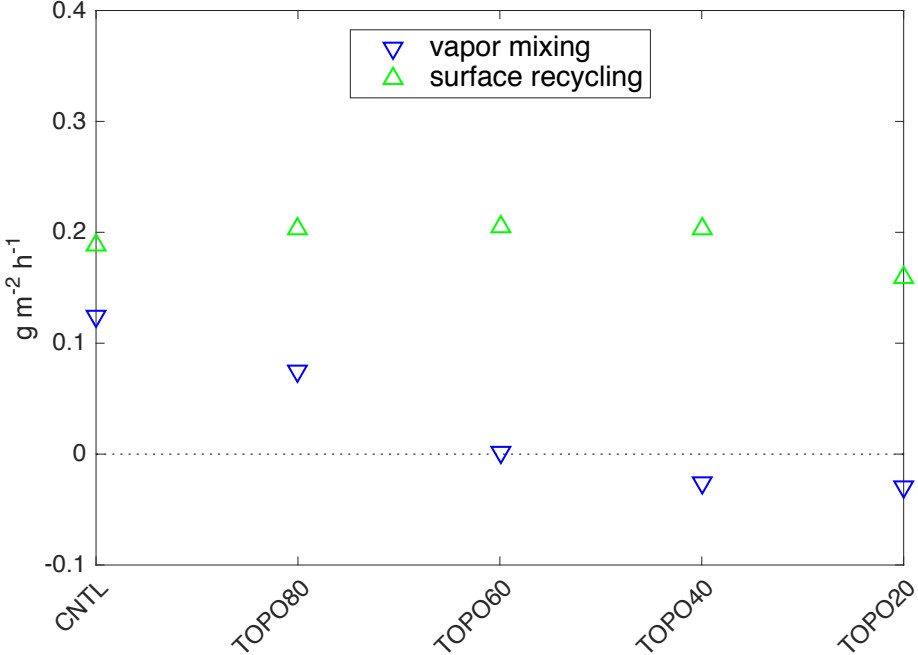

**Figure 12. The mass flux of $^{18}$O (g m$^{-2}$ h$^{-1}$)) from vapor mixing and surface recycling on the Tibetan Plateau. Note positive (negative) values indicate sources (sinks) of $^{18}$O. Surface recycling is the largest sources of $^{18}$O under all elevation scenarios. Vapor mixing is secondary as a source in high-elevation scenarios and becomes a sink in low-elevation scenarios.**



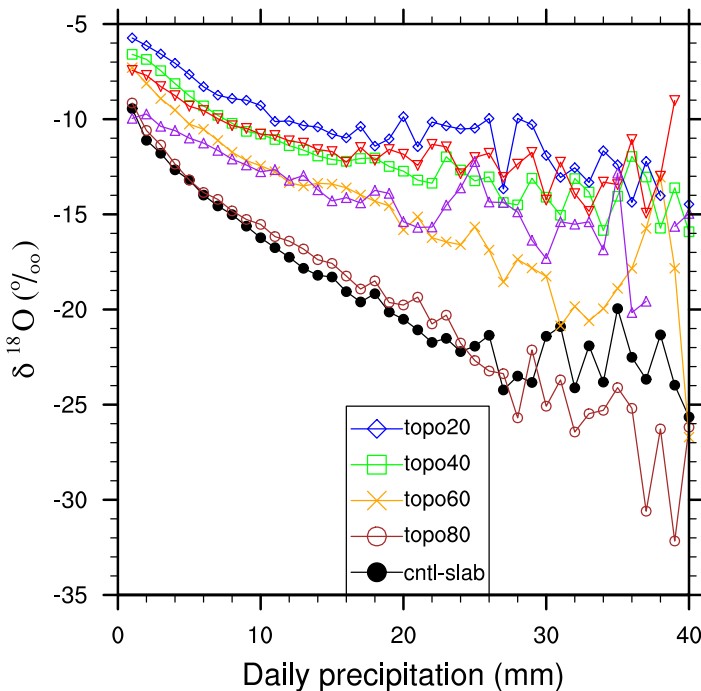

**Figure 13. ECHAM δ¹⁸Oₚ (‰) versus daily precipitation rate (mm/day) on the Tibetan plateau. ECHAM δ¹⁸Oₚ decreases with increasing precipitation under all elevation scenarios, but is greatest in high-elevation scenarios than in low-elevation scenarios.**



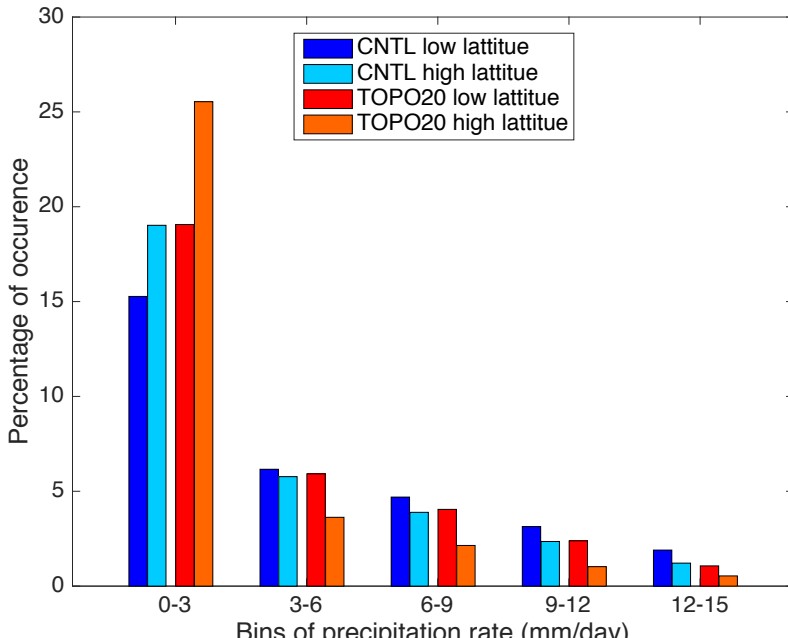

5    **Figure 14. Relative occurrence of (%) daily precipitation rates on the Tibetan Plateau for lower latitudes (averaged over 30~32° N, 85~100° E) higher latitudes (averaged over 34~36° N, 85~100° E). Only two elevation scenarios (CNTL and TOPO20) are shown here since other scenarios are identical to these two. Low precipitation rate events are more frequent at higher latitudes than at lower latitudes under all elevation scenarios. See Figure 13 for a demonstration of how this rainfall distribution impacts $\delta^{18}O_p$.**





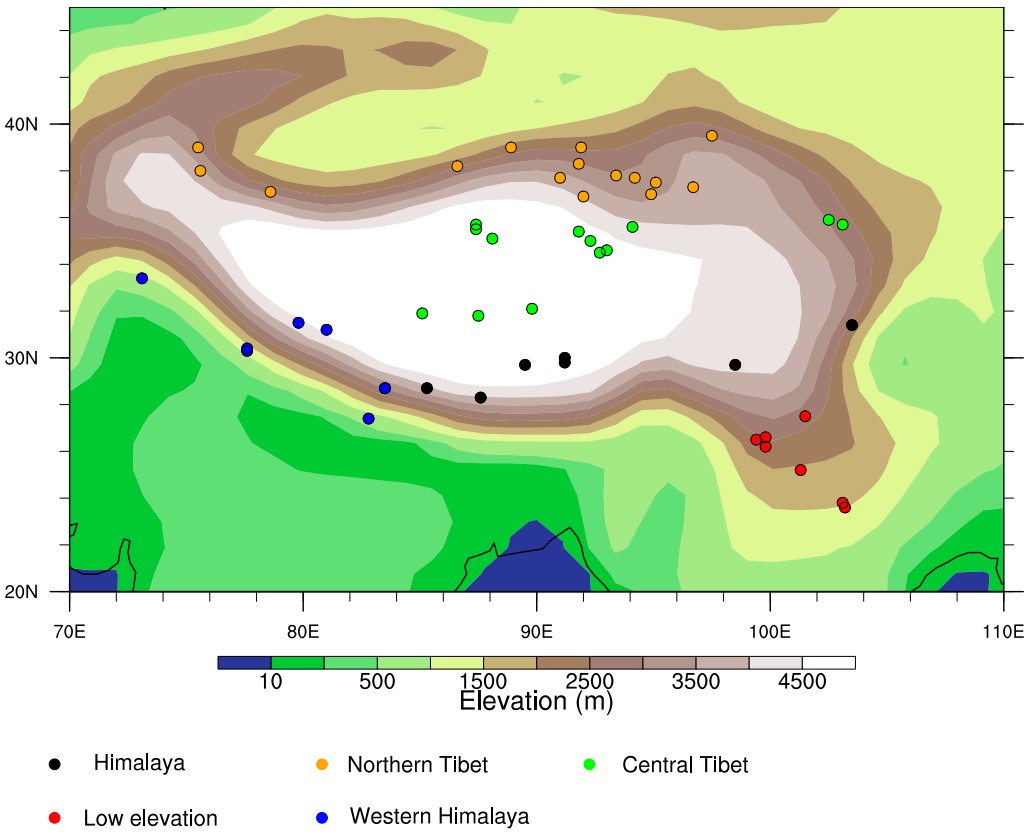

**Figure 15. Map of $\delta^{18}O_p$-paleoaltimetry sites (filled circles) plotted on surface elevation (shaded). Sites are classified by their type (see Section 4.2 in the paper for more details on the categories and Table S1 for a list of the sites).**

