# Peer review of "Precipitation $\delta^{18}O$ on the Himalaya-Tibet orogeny and its relationship to surface elevation"

_Climate of the Past, 2018_

## Referee Comment (RC1) · A. Licht (Referee) · 21 Oct 2018

The paper provides results from climate simulations with a water-isotope module to study the stability of rainfall isotopic lapse rates across the Tibetan-Himalayan orogen with varying altitude. The potential implications of this study are important, because the stability of these isotopic lapse rates through time and through different stages of uplift is a common assumption made by Tibetan paleoaltimetry studies and has been virtually unverified.

I must first say that I am very sympathetic with the effort made by the authors to investigate the behavior of these lapse rates with varying geography. This study is definitely

a great contribution to paleoaltimetry – so much has been written on the paleotopography of the Tibetan Plateau without a clear understanding of atmospheric and water isotopic dynamics of South Asia. I am not a climate modeler –I am from the data side and I have been working lately on paleoaltimetry topics; yet the writing is clear and the interpretations are understandable for non-modelers, ensuring that the paper will have a real impact on paleoaltimetry practices.

The manuscript is very well-written; the interpretations are reasonable in the light of the provided climate simulations. I have little to say and I just have two main concerns:

1) First, I am amazed about the amount of misfit between the Control experiment and modern data when it comes to rainfall d18O, particularly on the Tibetan Plateau itself. 2-5 permil of misfit in central Tibet is huge, when the variation from the Himalayan foothills to the top of the Plateau is of ∼10 permil. The source of the mismatch is stated as "unclear" (page 9, line 22), which is not satisfactory as Tibet is the area of interest. Please discuss this in more details and explain what the model could do wrong –explaining the mismatch with the interannual variability in rainfall d18O measurements is not satisfactory neither. Also, compare with other models (is ECHAM the only model to do that? What does Botsuyn et al say about LMDz iso over Tibet?). This should be a main discussion point in section 4.3 (caveats).

2) There is no discussion about monsoonal run-off into the Bengal Bay and the related amplification of the rainfall isotopic depletion, which is known to be a key control on rainfall d18O in South Asia. An essential paper on this topic is missing from the bibliography: Breitenbach et al (2010 EPSL). Briefly, this paper shows how seasonal (monsoonal) run-off of isotopically-depleted water into the Bengal Bay result in water stratification in the Bengal Bay and higher than normal sea water d18O, that increase rainfall isotopic depletion during the late monsoon season and explain the lowest rainfall d18O values. This type of seasonal effect would have a huge impact on rainfall d18O over the orogen and could have changed significantly with paleotopography (and monsoonal intensity). It sounds to me that the simulations provided in this manuscript

do not take into account these game-changing seasonal effects. I would like to hear more about how is set up the lower boundary d18O values (page 4, line 20); it sounds essential. Are the lower boundary d18O values varying through the year, or set as constant for the entire year?

More minor comments:

-page 3 line 2: pedogenic and lacustrine carbonates.

-page 3 line 25: Quade et al (2011, AJS) instead of Bershaw et al (2012). (Quade and coauthors were the first).

-Page 7 line 29, page 8 line 1: "summer precipitation decreases from ..." Where? On the whole Tibet?

-page 8 line 19-21, page 9 line 4-5: Actually, the oldest loess deposits are now dated to the Eocene (see Licht et al., 2014, Nature; 2016, Nature Communications; Li et al., 2018, Nature Communications). Similarly, the onset of the modern EASM in the early Miocene is highly debated, it is likely much older (there is quantity of papers on the topic over the last 4 years). Better to remove these statements (or nuance them).

Overall it is an excellent manuscript and I am looking forward to see it published once my two main comments have been addressed.

Alexis Licht

---

## Referee Comment (RC2) · Anonymous Referee #2 · 29 Oct 2018

This manuscript uses a water-isotope enabled AGCM to explore the relationship between Himalaya-Tibet elevation and delta-18O in rainfall. Such studies are critical since stable isotope-based paleoaltimetry has been widely used in the last years to constrain the uplift history of orogens. The ms is clearly structured, mostly well-written and, despite some rare confusions & ambiguities, quite convincing. I think the ms could be published with some minor revisions that will improve the overall discussion.

The authors make use of sensitivity experiments to Tibet/Himalaya height to understand processes driving rainfall d18O. In that respect, this ms has a lot in common with a previous paper authors refer to, Botsyun et al. 2016. Results bring also a similar

message, i.e. that there are many processes determining the ultimate O18 composition of precipitation waters that stable-isotope-based paleoaltimetry do not account for. Despite these similarities, I think this ms is a very important contribution for the geosciences community. First, because comparable results obtained with different GCMs reinforce the previous findings by -obviously- making the overall message less model-dependent. Second because authors have added a 1-D model that helps explore the mechanisms sequentially, in a kind of complementary way.

Still the ms would benefit from a deeper discussion of the differences/similarities between these 2 studies, especially regarding 3 points : The role of relative humidity values for re-evaporation processes, the role of the "amount effect" and the sensitivity (or not) of the results to convective/large-scale precipitation partitioning in the models.

This latter point is the only "grey area" of the ms in my opinion, for which I think some clarifications are required. Specifically, in section 3.6, after quantifying the different sources of 18O mass fluxes, authors argue that the decrease in large-scale/convective rainfall ratio leads to Rayleigh distillation weakening in low-elevation scenarios (page 12). I agree with this statement, but I think the implications of convection regime on the isotopes should be better explained. This interpretation and subsequent discussion would actually benefit from clarifications about (i) how convection influence water isotopes in general, (ii) how ECHAMiso convective scheme deal with these processes, and (iii) how one can link that to the amount effect. I'd suggest these clarifications to be made as soon as the introduction. Some useful references for that could be Bony et al. (2008) and Risi et al. (2008) (JGR Atmospheres).

I also think the ms could be improved with a small sketch depicting the sequence of processes tested between sections 3.4 and 3.7.

A sort discussion about the use of fixed SSTs and the expected changes in results if a fully-coupled GCM was used (dynamic coupling between changes in elevation, SST responses and advection of moisture towards the area for example), although putative,

would be interesting to inspire future studies.

Lastly, this subject is very active, and new papers have been out in the meantime of this ms submission. For example, I think the discussion would benefit from the recent synthesis published by Rugenstein Caves & Chamberlain earlier this month (!) in Earth Science Reviews.

I summarize in the following a few remarks and minor concerns I had with the ms, trying to follow its chronology.

Introduction. Line 23: Regarding the impact of mountains on biogeography, there's a recent contribtion by Antonelli et al. in Nature Geosciences that would be worth citing (Nature Geosciencevolume 11, pages718–725 (2018)). Line 25: Raymo et al. (1988) is kind of outdated to explain uplift/CO2 links, especially since the studies showing the impacts of organic carbon burial in this drawdown (see for exemple Galy et al., Nature volume 450, pages 407–410, 2007 and/or Maffre et al. EPSL, 2018.

Methods. Line 16: I am a bit puzzled: If a slab ocean is used, then SSTs are not prescribed, but should be calculated via atmospheric heat fluxes and prescribed ocean heat fluxes. Or am I missing something ?

Page 6. Line 14. I think this use slopes ratio to decipher the relative effects of altitude/latitude in low-elevation simulations is unclear. Please reformulate.

Results. General remark. Authors should b cautious about the way they present results : It is not always clear if one deal with JJA results or annual-mean.

I think section 3.3, i.e. ECHAM-iso validation, should come first in the results section. The map of simulated O18 and actual datapoints should be moved from the Supplemental Material back to the main text.

Section 3.1 needs rewriting: First of all, authors need to homogenize the units used for discussion of precipitation rates. Sometimes it's mm/d, in some other places it's mm/y. Example : Figure 3. It deals with JJA rainfall, and units are in mm/yr, which

is quite confusing. I recommend to switch all rainfall amount results to mm/day in the entire ms. Authors also use Fig. 3 to discuss wind reversals and changes in latitudinal rainfall patterns in lowered topography scenarios, but the way 850 hPa streamlines are designed + the poor choice of the colorbar (white threshold at 1800 mm/yr) make it impossible for the reader to check authors statements. Two suggestions to deal with this issue : 1/ Refocus the region over the region of interest, change to mm/d and rainfall colorbar/threshold. 2/ Define some lat/lon sections to show the actual wind reversals, or just add the zero-line of zonal/meridional wind component on each map. I am not convinced by Fig. 4 either. IM decreases from 28.5 m.s-1 to 23, which is a 20% decrease. With an improved Fig 3. & a sentence stating that there's this 20% decrease in IM, the message will be clearer and authors will save space for a figure.

The discussion about the Himalayas impact on IM dynamics (3.2, page 8-9) and the long-standing debate about air-mass isolations versus thermal contrasts initiated by Boos and colleagues is interesting, but might be more relevant in the discussion part.

Moisture source influence. Page 9 : lines 28-30 should be moved back to model validation section. Page 10 line 9 to 14. Figure 7-8 do not show that d18O values of RDM and ECHAM are "close in all elevations scenarios." Actually, the slopes are similar but d18O are systematically shifted to lower values. Tables tend to cancel this signal by averaging over a box, but authors should moderate their statement.

Page 10, line 16: should read table 1 instead of table 3 ?

Fig. 7-8: Why does RDMfixed_T scenario from Table 1 not appear on these figures ?

Figure 13: Seven lines, but only 5 legends.

line 16 : "of" missing. Page 12, line 2 : "O18-enriched".
* * *

---

## Author Comment (AC1) · 12 Dec 2018

December 13, 2018

Dear Editor:

Thank you for the opportunity to revise our manuscript, "Precipitation $\delta^{18}O$ on the Himalaya-Tibet orogeny and its relationship to surface elevation" by Hong Shen and Christopher J. Poulsen submitted for publication in *Climate of the Past*. We found the reviewers' comments to be very constructive and think that they improve the quality of the manuscript.

Enclosed please find a revised version of the manuscript that incorporates the reviewers' suggestions. We also include our responses to the reviewers' comments, which describe in detail the changes that we have made in the revised manuscript.

Thanks again for your consideration of our manuscript. Please contact me with any questions.

Sincerely,
Hong Shen

**Response to Reviwer # 1 Dr. Alexis Licht:**

*The paper provides results from climate simulations with a water-isotope module to study the stability of rainfall isotopic lapse rates across the Tibetan-Himalayan orogen with varying altitude. The potential implications of this study are important, because the stability of these isotopic lapse rates through time and through different stages of uplift is a common assumption made by Tibetan paleoaltimetry studies and has been virtually unverified.*

*I must first say that I am very sympathetic with the effort made by the authors to investigate the behavior of these lapse rates with varying geography. This study is definitely a great contribution to paleoaltimetry – so much has been written on the paleotopography of the Tibetan Plateau without a clear understanding of atmospheric and water isotopic dynamics of South Asia. I am not a climate modeler –I am from the data side and I have been working lately on paleoaltimetry topics; yet the writing is clear and the interpretations are understandable for non-modelers, ensuring that the paper will have a real impact on paleoaltimetry practices.*

*The manuscript is very well-written; the interpretations are reasonable in the light of the provided climate simulations. I have little to say and I just have two main concerns:*

*1) First, I am amazed about the amount of misfit between the Control experiment and modern data when it comes to rainfall d18O, particularly on the Tibetan Plateau itself. 2-5 permil of misfit in central Tibet is huge, when the variation from the Himalayan foothills to the top of the Plateau is of _10 permil. The source of the mismatch is stated as "unclear" (page 9, line 22), which is not satisfactory as Tibet is the area of interest. Please discuss this in more details and explain what the model could do wrong –explaining the mismatch with the interannual variability in rainfall d18O measurements is not satisfactory neither. Also, compare with other models (is ECHAM the only model to do that?*

To address the reviewer's concern, we have added in Section 3.1 (see page 8, lines 15-19) that another source for this mismatch could be that the simulated precipitation is higher than the observed. Over the region of east-central Tibet (89° E-102° E, 32° N-35° N), the JJA precipitation rate in ECHAM5 is 3.9 mm/day, compared to 2.8 mm/day in CMAP. The high precipitation rate in ECHAM could further deplete raindrops through the amount effect, which is shown to be an important factor controlling $\delta^{18}O_p$ on the Tibetan Plateau in Section 3.6.

*What does Botsuyn et al say about LMDz iso over Tibet?). This should be a main discussion point in section 4.3 (caveats).*

As reported in Botsuyn et al., the LMDziso model also simulated rainfall $\delta^{18}O$ over the central Tibetan Plateau that are too low (by 1-4‰) (See Page 8, lines 9-11 and lines 16-19).

*2) There is no discussion about monsoonal run-off into the Bengal Bay and the related amplification of the rainfall isotopic depletion, which is known to be a key control on rainfall d18O in South Asia. An essential paper on this topic is missing from the bibliography: Breitenbach et al (2010 EPSL). Briefly, this paper shows how seasonal (monsoonal) run-off of isotopically-depleted water into the Bengal Bay result in water stratification in the Bengal Bay and higher than normal sea water d18O, that increase rainfall isotopic depletion during the late monsoon season and explain the lowest rainfall d18O values. This type of seasonal effect would have a huge impact on rainfall d18O over the orogen and could have changed significantly with paleotopography (and monsoonal intensity). It sounds to me that the simulations provided in this manuscript do not take into account these game-changing seasonal effects.*

The reviewer is correct that ECHAM does not account for these variations in seawater $\delta^{18}O$. However, variations in Bay of Bengal seawater $\delta^{18}O$ have a minimal impact on the isotopic composition of vapor evaporated from the surface. We show this using the Craig-Gordon model with inputs of monthly SST, air temperature and relative humidity from our ECHAM5 CNTL simulation, and monthly seawater $\delta^{18}O$ from Breitenbach et al (2010). As calculated by the C-G model, the annual

range of vapor $\delta^{18}O$ evaporated from the ocean surface would be 0.24‰. This small range is not surprising since seawater $\delta^{18}O$ varies by only 2.5‰ under modern conditions (Breitenbach et al., 2010). We speculate that the seawater $\delta^{18}O$ range of the Bay of Bengal may have been even smaller if the Asian monsoon were weaker under reduced elevation scenarios.

During past times when $pCO_2$ was higher and the seasonal variation of seawater $\delta^{18}O$ were higher (for example, 4.5‰ as in Breitenbach et al.), the vapor $\delta^{18}O$ range is likely to be slightly larger, but still negligible. To show this, we increased the air temperature and SST from the CNTL simulation by 4 °C according to a recent simulation done by Vahlenkamp et al. (2018, EPSL) with a $pCO_2$ of 1000ppm. As a result of the temperature increase, the range of vapor $\delta^{18}O$ evaporated from the surface is less than 1‰. These estimates illustrate that seasonal variations in seawater $\delta^{18}O$ would have a very small influence on rainfall $\delta^{18}O$. We have added this point to our manuscript in Section 4.3 (see page 17, lines 21-26).

*I would like to hear more about how is set up the lower boundary d18O values (page 4, line 20); it sounds essential. Are the lower boundary d18O values varying through the year, or set as constant for the entire year?*

We thank the reviewer for pointing out this omission in our description of the boundary conditions. We have added text to Section 2.1 (see Page 4, lines 18-19) describing our prescription of seawater $\delta^{18}O$, which follows the method used by Li et al. (2016). In short, we used the observed annual mean seawater $\delta^{18}O$ by LeGrande and Schmidt (2006), meaning there is no seasonal change in seawater $\delta^{18}O$.

*More minor comments:*
*-page 3 line 2: pedogenic and lacustrine carbonates.* Changed.

*-page 3 line 25: Quade et al (2011, AJS) instead of Bershaw et al (2012). (Quade and coauthors were the first).* Changed.

*-Page 7 line 29, page 8 line 1: "summer precipitation decreases from . . ." Where? On the whole Tibet?*

We have modified the text to indicate the region. The sentence reads: "summer precipitation decreases from ~3 mm day$^{-1}$ in CNTL to ~0.1mm day$^{-1}$ on the western Himalayan slope in TOPO20."

*-page 8 line 19-21, page 9 line 4-5: Actually, the oldest loess deposits are now dated to the Eocene (see Licht et al., 2014, Nature; 2016, Nature Communications; Li et al., 2018, Nature Communications). Similarly, the onset of the modern EASM in the early Miocene is highly debated, it is likely much older (there is quantity of papers on the topic over the last 4 years). Better to remove these statements (or nuance them).*

We thank the reviewer for clarifying this. In response, we have deleted lines 19-21 on p.8 in the original version and have added text on page 10 (lines 4-8) in this version indicating that the onset of the EASM is highly debated and $pCO_2$ is an important factor.

*Overall it is an excellent manuscript and I am looking forward to see it published once my two main comments have been addressed.*

**Response to Reviwer # 2:**

*This manuscript uses a water-isotope enabled AGCM to explore the relationship between Himalaya-Tibet elevation and delta-18O in rainfall. Such studies are critical since stable isotope-based paleoaltimetry has been widely used in the last years to constrain the uplift history of orogens. The ms is clearly structured, mostly well-written and, despite some rare confusions & ambiguities, quite convincing. I think the ms could be published with some minor revisions that will improve the overall discussion.*

*The authors make use of sensitivity experiments to Tibet/Himalaya height to understand processes driving rainfall d18O. In that respect, this ms has a lot in common with a previous paper authors refer to, Botsyun et al. 2016. Results bring also a similar message, i.e. that there are many processes determining the ultimate O18 composition of precipitation waters that stable-isotope-based paleoaltimetry do not account for. Despite these similarities, I think this ms is a very important contribution for the geosciences community. First, because comparable results obtained with different GCMs reinforce the previous findings by -obviously- making the overall message less modeldependent. Second because authors have added a 1-D model that helps explore the mechanisms sequentially, in a kind of complementary way.*

*Still the ms would benefit from a deeper discussion of the differences/similarities between these 2 studies, especially regarding 3 points : The role of relative humidity values for re-evaporation processes, the role of the "amount effect" and the sensitivity (or not) of the results to convective/large-scale precipitation partitioning in the models.*

*This latter point is the only "grey area" of the ms in my opinion, for which I think some clarifications are required. Specifically, in section 3.6, after quantifying the different sources of 18O mass fluxes, authors argue that the decrease in large-scale/convective rainfall ratio leads to Rayleigh distillation weakening in low-elevation scenarios (page 12). I agree with this statement, but I think the implications of convection regime on the isotopes should be better explained. This interpretation and subsequent discussion would actually benefit from clarifications about (i) how convection influence water isotopes in general, (ii) how ECHAMiso convective scheme deal with these processes, and (iii) how one can link that to the amount effect. I'd suggest these clarifications to be made as soon as the introduction. Some useful references for that could be Bony et al. (2008) and Risi et al. (2008) (JGR Atmospheres).*

We agree that some additional discussion of how convection affects $\delta^{18}O_p$ would improve the manuscript. Convection influences $\delta^{18}O_p$ in two ways: (i) by shifting below cloud-base fractionation from mostly equilibrium to kinetic fractionation, mostly through sub-cloud evaporation (as in Risi et al., 2008), and (ii) by enhancing the updraft of $\delta^{18}O$-enriched vapor from the boundary layer to upper levels (as in Bony et al. 2008).

Bony et al. 2008 highlighted convective updraft as the main factor that enriches $\delta^{18}O_p$ in convective scheme. Our methods and experimental design are different from Bony et al. 2008, in which they used a single column model without horizontal base flow, and thus, they can easily account for the updraft by advection. In our model, in the Himalayas, the upslope flow is both in horizonal and vertical direction, and the upslope flow changes as elevation is reduced. So, the convective updraft in our simulations are included in the advection term (i.e. vapor mixing in the manuscript). Similarly, the amount effect is included mostly in the fluxes of convective rain in 3.6..

To further address the reviewer's point, firstly, we have addressed in section 2.3 (page 7 lines 18-20) that not being able to isolate these individual effects is a limitation of our method and suggest this as a goal of future studies. Secondly, we added in Methods (2.1, page 4 lines 23-26) a description of how relative humidity is linked to convective scheme and sub-cloud re-evaporation, and how ECHAM deals with convective rain and contributes to $\delta^{18}O_p$ via the amount effect in section 3.6 (page 12, lines 27-30). The additional text in the section reads:
 "The increase in convective rainfall in these cases leads to greater kinetic fractionation through sub-cloud evaporation of falling rain, which is only partially equilibrated with the surrounding vapor (see 2.1), and an enrichment in the isotopic composition of rain. The RDM does not capture this enrichment because it does not include sub-cloud evaporation."

*I also think the ms could be improved with a small sketch depicting the sequence of processes tested between sections 3.4 and 3.7.*

We appreciate the reviewer's suggestion and have added a cartoon sketch (Fig. 17) to show the dominant processes for high and low elevation scenarios.

*A sort discussion about the use of fixed SSTs and the expected changes in results if a fully-coupled GCM was used (dynamic coupling between changes in elevation, SST responses and advection of moisture towards the area for example), although putative, would be interesting to inspire future studies.*

We agree that this is a point worth further discussion. We have added a short discussion in Section 4.3 (page 17, lines 13-20) that reads:
"Another limitation of our modelling strategy is our use of a slab ocean model, which does not account for ocean circulation changes that would result from the changes in topography that we prescribe. To the best of our knowledge, no study has specifically investigated the response to a reduction in the elevations of the Himalayas and the Tibetan Plateau. It is not hard to imagine regional sea surface changes that might influence inland precipitation. For example, we speculate that under lower elevations and weaker monsoon winds, ocean upwelling along the western coast of Bay of Bengal and the Arabian Sea would be reduced, leading to higher sea-surface temperatures (SSTs). In a study of the East Asian response to historical SST warming, higher SSTs led to greater precipitation over the Indian Ocean and Pacific Ocean due to enhanced local convection and less precipitation along the Himalayan front, and further weakening of the Asian monsoon (Li et al., 2010)"

*Lastly, this subject is very active, and new papers have been out in the meantime of this ms submission. For example, I think the discussion would benefit from the recent synthesis published by Rugenstein Caves & Chamberlain earlier this month (!) in Earth Science Reviews.*

Yes, there are some new papers on this topic. We added the Rugenstain paper in introduction (page 3, line 27) to address the idea that the south-to north gradient of $\delta^{18}O$ is consistent since the early Eocene, and also in 3.3 (page 10, line 6) to address the influence of $PCO_2$ on EASM. We also include a new paper (Tabor et al, 2018) in 4.3 (page 17, line 32) to address the influence of orbital parameters.

*I summarize in the following a few remarks and minor concerns I had with the ms, trying to follow its chronology.*

*Introduction. Line 23: Regarding the impact of mountains on biogeography, there's a recent contribtion by Antonelli et al. in Nature Geosciences that would be worth citing (Nature Geoscience volume 11, pages718–725 (2018)). Line 25: Raymo et al. (1988) is kind of outdated to explain uplift/CO2 links, especially since the studies showing the impacts of organic carbon burial in this drawdown (see for exemple Galy et al., Nature volume 450, pages 407–410, 2007 and/or Maffre et al. EPSL, 2018.*

We have incorporated the reviewer's reference suggestions.

*Methods. Line 16: I am a bit puzzled: If a slab ocean is used, then SSTs are not prescribed, but should be calculated via atmospheric heat fluxes and prescribed ocean heat fluxes. Or am I missing something ?*

We thank the reviewer for catching this inconsistency and have corrected it in the text. SSTs are calculated using a slab ocean (page 4, line 17).

*Page 6. Line 14. I think this use slopes ratio to decipher the relative effects of altitude/latitude in low-elevation simulations is unclear. Please reformulate.*

We added that we used this slope ratio because the lapse rate of $\delta^{18}O$ with elevation could also be partly attributed to the continental effect, rather than the Rayleigh distillation process (page 6, lines 17-18). Thus, this ratio shows how strong/weak Rayleigh distillation is especially in low-elevation scenarios.

*Results. General remark. Authors should b cautious about the way they present results: It is not always clear if one deal with JJA results or annual-mean.*

We thank the reviewer for this comment. In addition to stating in the Methods (page 5, lines 10-12) that only JJA climate variables and annual-mean precipitation-weighted $\delta^{18}O_p$ are analyzed, we have specified this in other locations throughout the manuscript, especially in the figure captions.

*I think section 3.3, i.e. ECHAM-iso validation, should come first in the results section. The map of simulated O18 and actual datapoints should be moved from the Supplemental Material back to the main text.*

We agree and have made these changes. The previous Section 3.3 is now Section 3.1. The map of simulated $\delta^{18}O$ is now Fig. 5

*Section 3.1 needs rewriting: First of all, authors need to homogenize the units used for discussion of precipitation rates. Sometimes it's mm/d, in some other places it's mm/y. Example : Figure 3. It deals with JJA rainfall, and units are in mm/yr, which is quite confusing. I recommend to switch all rainfall amount results to mm/day in the entire ms. Authors also use Fig. 3 to discuss wind reversals and changes in latitudinal rainfall patterns in lowered topography scenarios, but the way 850 hPa streamlines are designed + the poor choice of the colorbar (white threshold at 1800 mm/yr) make it impossible for the reader to check authors statements. Two suggestions to deal with this issue : 1/ Refocus the region over the region of interest, change to mm/d and rainfall colorbar/threshold. 2/ Define some lat/lon sections to show the actual wind reversals, or just add the zero-line of zonal/meridional wind component on each map.*

We agree with the reviewer that it is better to change all units to mm/day. We changed it both in Fig. 3 (Fig. 6 in this version) and in the text (see 3.2). Then we also changed the choice of the color bar for precipitation, and marked the wind reversal.

*I am not convinced by Fig. 4 either. IM decreases from 28.5 m.s-1 to 23, which is a 20% decrease. With an improved Fig 3. & a sentence stating that there's this 20% decrease in IM, the message will be clearer and authors will save space for a figure.*

We also used Fig.4 (Fig. 7 in this version) to show the threshold between 60% and 40% of modern height, but did not make it clear enough. We further clarified in the next sentence that we are getting this threshold from this figure and we think it better to keep this figure (page 9, lines 12-14).

*The discussion about the Himalayas impact on IM dynamics (3.2, page 8-9) and the long-standing debate about air-mass isolations versus thermal contrasts initiated by Boos and colleagues is interesting, but might be more relevant in the discussion part.*

We respectfully disagree with the reviewer on this point. Firstly, we think that it flows well to clarify on the IM dynamics directly after showing the results from TOPO20a and TOPO20b. Secondly, the IM dynamics is very short and fits better in the results part than in the discussion part.

*Moisture source influence. Page 9 : lines 28-30 should be moved back to model validation section.*

We agree with this point and moved the sentence up to page. 7, lines 22-24.

*Page 10 line 9 to 14. Figure 7-8 do not show that d18O values of RDM and ECHAM are "close in all elevations scenarios." Actually, the slopes are similar but d18O are systematically shifted to lower values. Tables tend to cancel this signal by averaging over a box, but authors should moderate their statement.*

We think that the reviewer might be looking at the RDM-fixed moisture (diamond) instead of the ECHAM (triangle), since the diamond was in red and more eye-catching. We switched the colors of the diamonds with the triangle to help avoid this.

*Page 10, line 16: should read table 1 instead of table 3 ?*

This is table 1, since table 1 shows different moisture source scenarios with fixed T or ECHAM T and ECHAM RH.

*Fig. 7-8: Why does RDMfixed_T scenario from Table 1 not appear on these figures ?*

The $RDM_{Fixed\_T}$ scenario was introduced on the purpose of accounting for individual contributions from either T or RH change with reduced elevation. To initiate the RDM in reconstructing past elevation, people most commonly used fixed T and RH (as in $RDM_{Fixed}$) rather than fixed T and changing RH (as in $RDM_{Fixed\_T}$), and this is why we added only $RDM_{Fixed}$ in Fig 7-8 for comparison.

*Figure 13: Seven lines, but only 5 legends.*

Removed the two additional lines that is not discussed in the text.

line 16 : "of" missing.
The reviewer didn't indicate the page number. We have re-read the paper and made minor edits throughout. We may have caught this missing "of" in that process.

Page 12, line 2 : "O18-enriched". Changed.

[revised manuscript text omitted]